# Beyond Straight Paths: Erasure-Redraw Sampling for Diverse Flow Matching

## Abstract

While Flow Matching models have achieved state-of-the-art performance, their reliance on deterministic, straight-path ODE sampling limits their capacity to explore the multi-modal nature of data distributions under linguistic constraints. For example, a prompt for a "robot" may encompass distinct semantic modes (*e.g.*, "red" vs. "yellow"), yet deterministic solvers often collapse into a single interpretation. This limitation is particularly restrictive in interactive scenarios where users desire to "redraw" specific regions—exploring diverse local alternatives while following the same prompt and global context constraints. To bridge this gap, we propose Erasure-Redraw Sampling, a training-free framework that enables high-quality local semantic variations via a zigzag (backward-and-forward) sampling trajectory. Our method alternates between two phases: 1, Erasure: stochastic prompts are introduced during backward sampling to trigger mode-switching by effectively clearing existing local details. 2, Redraw: visual prompts serve a dual purpose—guiding the synthesis of new semantic details while enforcing spatial coherence during a forward pass. Experimental results demonstrate that our method effectively balances global consistency with local multi-modality, offering a robust, plug-and-play solution for diverse generation.

## 1. Introduction

A fundamental challenge in generative learning frameworks is known as the "generative learning trilemma" (Xiao et al., 2022), which posits that models struggle to achieve three crucial goals: high sample quality, fast sampling, and mode coverage (diversity). While recent state-of-the-art methods like GAN (Iglesias et al., 2023), Diffusion Models (Chan,

Prompt: *"a small robot on the wooden table"*

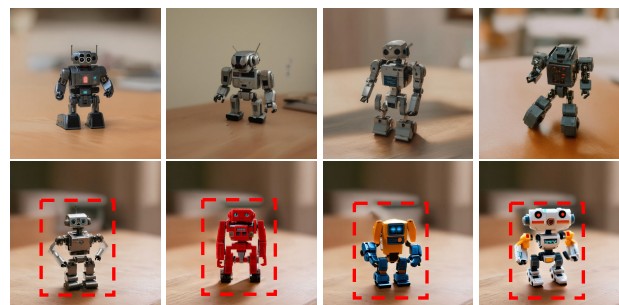

*Figure 1.* Local Diversity Demonstration. Our approach enables the synthesis of diverse variations. The top row illustrates the baseline (SD 3.5) across multiple sampling attempts, which exhibit mode collapse with highly redundant content. The bottom row, with the target region marked by a red boundary, showcases our method's ability to inject localized diversity while preserving background structure.

2024) and Flow Matching (Lipman et al., 2024) have made significant strides in achieving high quality and speed, the explicit enhancement of diversity remains a significant and under-explored challenge.

Due to limited mode coverage, generating diverse image variations typically requires a prohibitively large sampling budget. Consider the scene in Figure 1 (first row), where the textual prompt pt = "a small robot on a wooden table." Across different random seeds, the generations converge to nearly identical robotic designs. Even attributes not explicitly constrained by the prompt, such as color and shape, fail to exhibit meaningful variation.

This inefficiency critically hinders interactive applications. In such scenarios, instead of requiring an entirely new image with each interaction, users only want to explore only local diversity—varying specific aspects such as an object's color or shape. Thus, there is a critical need for a sampling mechanism that can explore diverse semantic interpretations within the specific region under the same prompt constraint.

Current sampling techniques struggle to navigate local semantic modes when restricted to straight ODE paths. The most intuitive solution—sampling independent and identically distributed (I.I.D) noise from a Gaussian prior—is highly sample-inefficient. This "trial-and-error" approach

---

[1]Anonymous Institution, Anonymous City, Anonymous Region, Anonymous Country. Correspondence to: Anonymous Author <anon.email@domain.com>.

Preliminary work. Under review by the International Conference on Machine Learning (ICML). Do not distribute.

often yields redundant results and lacks the precision required to explore specific local modes. Alternatively, modifying the prompt embedding (Ruan et al., 2025; Meng et al., 2025) frequently triggers unintended holistic shifts, due to the global influence of cross-attention mechanisms. Similarly, while joint sampling strategies (Corso et al., 2024; Liu et al., 2025; Morshed & Boddeti, 2025) remain computationally exhaustive and struggle to preserve the established global structural context.

To bridge this gap, we propose *Erasure-Redraw Sampling* to generate diverse local variations while preserving the original random seed and textual conditioning. Our approach implements a stochastic trajectory perturbation using a *zigzag sampling path*. This strategy alternates between stochastic backward steps to dissolve local details and guided forward steps to synthesize novel attributes while enforcing spatial coherence. Intuitively, the backward step acts as a "erasure" of context mirroring previous results, while the forward step serves as a generative "redraw" phase to diverge from the prior states. This Erasure-Redraw (backward-and-forward) Sequence enables the model to escape local modes and explore adjacent high-probability regions of the data manifold. Consequently, our method ensures diversity in modified regions while anchoring the global composition to the original structural prior as shown in Figure 1 (second row).

The visual prompt employed in the forward process is formulated as $pv_0 + \sum_{i=1} pv_i$. Within this composition, $pv_0$ serves as a structural constraint to ensure the context in the unmasked regions remains unchanged, while each $pv_i$ represents a repulsive objective designed to deviate the generated content from the $i$-th iteration within the masked region. To inject stochasticity into the otherwise deterministic inference process, stochastic prompts are constructed by appending random words to the original textual prompt pt. These augmented prompts guide the reverse inference phase, effectively seeking a novel initialization point that share the same umask region and have diverse illustration for the mask region for the subsequent forward process. In summary, the contributions of our work are as follows:

1. **Zigzag Trajectory:** We introduce a sampling strategy that moves beyond straight-path ODEs, strategically alternating between diversity-seeking perturbations and structural preservation.

2. **Sequential Generation:** Unlike batch-heavy joint sampling, our framework enables an efficient *sequential* mechanism, uniquely optimized for interactive and iterative image refinement.

3. **High-Fidelity Diversity:** Empirical results demonstrate that our method achieves superior mode coverage and high-fidelity variations with lower computational overhead.

## 2. Related Work

Despite faster inference, achieving diverse mode coverage via independent trajectories remains a bottleneck. Redundant exploration of high-probability modes limits sample efficiency. This section reviews relevant literature.

**Noise Sampling:** To enhance diversity and efficiency, Corso et al. (2024) introduce particle guidance via joint-particle potentials, while Liu et al. (2025) propose a non-IID sampling framework for salient region coverage. In addition, some strategies optimize initialization by refining noise vector selection (Guo et al., 2024).

**Prompts Sampling:** To enhance output variations, Yun et al. (2025) and Um & Ye (2025) optimize prompt sampling, while Zhao et al. (2025) learn a soft-prompt distribution to capture broader diversity. Complementing these approaches, Ruan et al. (2025) model prompt embeddings via a Mixture-of-Gaussians to maintain semantic integrity across diverse samples. Alternatively, Miao et al. (2024) and Dombrowski et al. (2025) incorporate diversity losses and modules as substitutes for prompt-based diversification.

**Joint Sampling:** These methods optimize set-wise diversity within fixed sampling budgets. Morshed & Boddeti (2025) utilize Determinantal Point Processes to couple samples, while Kirchhof et al. (2025) integrate repulsion terms into the diffusion SDE to steer trajectories away from reference sets. Additionally, Parmar et al. (2026) propose a scalable group inference framework to simultaneously enhance group-wise diversity and visual quality. More recently, Kim et al. (2026) introduce Contrastive Noise Optimization, a straightforward yet effective strategy to mitigate mode collapse through contrastive objectives.

Despite the impressive generative performance, flow matching models (Lipman et al., 2024) often suffer from semantic misalignment or global inconsistency. These issues arise when the denoising process converges to local optima.

To mitigate this, existing works reinforce conditional signals via classifier-free guidance (Ho & Salimans, 2021), repeated conditional denoising (Lugmayr et al., 2022), or unconditional inversion (LiChen et al., 2025). More recently, test-time scaling methods (Ma et al., 2025; He et al., 2025; Zhang et al., 2025) have introduced stochasticity—such as mutating latent states or backtracking to previous timesteps—to escape local maxima under the guidance of reward models.

We observe that low diversity results from trajectories converging to stagnant local optima. To address this, we integrate *zigzag sampling* (LiChen et al., 2025) and *stochastic mutation* (Ma et al., 2025) into *Erasure-Redraw Sampling*. By avoiding deterministic, straight-line paths, our method facilitates robust mode exploration and enhances sample diversity.

## 3. Erasure-Redraw Sampling

Our method leverages visual and stochastic prompts within the Flow Matching model. Using a piecewise constant velocity field, we implement a zigzag inference path: stochastic prompts dissolve local modes during backward steps, while visual prompts guide the forward redrawing phase.

### 3.1. Piecewise Constant Velocity Field

Inference in Flow Matching is formulated as solving a probability flow ODE. Given a learned velocity field $v_{\boldsymbol{\theta}}(\boldsymbol{x}, \text{pt}, t)$, a sample $\boldsymbol{x}_0 \sim p_0(\boldsymbol{x})$ is transformed into $\boldsymbol{x}_1 \sim p_1(\boldsymbol{x})$ by solving the following initial value problem:

$$\frac{d\boldsymbol{x}_t}{dt} = \boldsymbol{v_\theta}(\boldsymbol{x}_t, \text{pt}, t), \quad t \in [0, 1], \tag{1}$$

subject to the initial condition $\boldsymbol{x}_{t=0} = \boldsymbol{x}_0$. Here, pt denotes the textual prompt specifying the semantic content of the generation, and $\boldsymbol{x}_0$ represents the initial noise.

While the ODE defines a continuous trajectory, numerical inference requires discretization. To this end, we adopt a *piecewise constant velocity field* approximation. We partition the interval $t \in [0, 1]$ into $N$ discrete steps. Within each sub-interval $[t_n, t_{n+1}]$ with step size $\Delta t = t_{n+1} - t_n$, we assume the velocity field is locally constant as $\boldsymbol{v}_n(\boldsymbol{x}_{t_n}, \text{pt})$. In the forward process, the state $\boldsymbol{x}$ then evolves from noise $\boldsymbol{x}_0$ to image $\boldsymbol{x}_1$ according to the discrete mapping $\phi : \mathcal{X} \to \mathcal{X}$:

$$\boldsymbol{x}_{t_{n+1}} = \phi(\boldsymbol{x}_{t_n}) = \boldsymbol{x}_{t_n} + \boldsymbol{v}_n^f(\boldsymbol{x}_{t_n}, \text{pt})\Delta t, \tag{2}$$

where $\boldsymbol{v}_n^f(\boldsymbol{x}_{t_n}) = \boldsymbol{v}_n(\boldsymbol{x}_{t_n}, \text{pt})$. Conversely, by defining the backward velocity as $\boldsymbol{v}_{n+1}^b(\boldsymbol{x}_{t_{n+1}}, \text{pt}) = \boldsymbol{v}_{n+1}(\boldsymbol{x}_{t_{n+1}}, \text{pt})$, the backward process follows the discrete mapping $\psi : \mathcal{X} \to \mathcal{X}$, enabling trajectory reversal:

$$\boldsymbol{x}_{t_n} = \psi(\boldsymbol{x}_{t_{n+1}}) = \boldsymbol{x}_{t_{n+1}} - \boldsymbol{v}_{n+1}^b(\boldsymbol{x}_{t_{n+1}}, \text{pt})\Delta t, \tag{3}$$

According to the *Change of Variables* formula, the generation density evolution under this mapping satisfies:

$$p_{n+1}(\boldsymbol{x}_{t_{n+1}}) = p_n(\boldsymbol{x}_n) \cdot |\det \nabla_{\boldsymbol{x}_n}\phi(\boldsymbol{x}_n)|^{-1}, \tag{4}$$

where the Jacobian matrix $\nabla_{\boldsymbol{x}}\phi(\boldsymbol{x}) = \frac{\partial(\boldsymbol{x}+\hat{\boldsymbol{v}}_n(\boldsymbol{x}, \text{pt})\Delta t)}{\partial \boldsymbol{x}} = \boldsymbol{I} + \Delta t \nabla_{\boldsymbol{x}}\hat{\boldsymbol{v}}_n(\boldsymbol{x}, \text{pt})$ describes the local geometry of the forward process. To introduce controlled diversity or guidance, we define a reshaped distribution $\hat{p}_n(\boldsymbol{x})$ modulated by an energy function $E_n(\boldsymbol{x})$ such that:

$$\hat{p}_n(\boldsymbol{x}) = \frac{1}{Z_n} p_n(\boldsymbol{x}) \exp(E_n(\boldsymbol{x})), \tag{5}$$

where $Z_n$ is the partition function. The following theorem characterizes how this energy-based reshaping affects the transition dynamics.

**Theorem 3.1** (Proof is left in the supplementary material). *The transition intensity of the reshaped distribution $\hat{p}_n$ can be expressed as a reweighting of the original transition intensity by the exponential change in energy:*

$$\hat{p}_n(\boldsymbol{x}_{t_{n+1}}|\boldsymbol{x}_{t_n}) = p_n(\boldsymbol{x}_{t_{n+1}}|\boldsymbol{x}_{t_n}) \exp(E_n(\boldsymbol{x}_{t_{n+1}}) - E_n(\boldsymbol{x}_{t_n})),$$

The modified velocity field $\hat{\boldsymbol{v}}_n(\boldsymbol{x})$ that yields the reshaped transition $\hat{p}(\boldsymbol{x}_{t_{n+1}}|\boldsymbol{x}_{t_n})$ is given by the following theorem.

**Theorem 3.2** (Proof is left in the supplementary material). *The velocity field $\hat{\boldsymbol{v}}_n^f(\boldsymbol{x}_{t_n}, \text{pt})$ that yields the reshaped transition $\hat{p}(\boldsymbol{x}_{t_{n+1}}|\boldsymbol{x}_{t_n}) = p(\boldsymbol{x}_{t_{n+1}}|\boldsymbol{x}_{t_n}) \exp(E_n(\boldsymbol{x}_{t_{n+1}}) - E_n(\boldsymbol{x}_{t_n}))$ is established by the following theorem.*

$$\hat{\boldsymbol{v}}_n^f(\boldsymbol{x}_{t_n}, \text{pt}) = \boldsymbol{v}_n^f(\boldsymbol{x}_{t_n}, \text{pt}) + \eta \nabla_{\boldsymbol{x}} E_n(\boldsymbol{x}_{t_n}) + \mathcal{O}(\Delta t), \tag{6}$$

### 3.2. Visual Prompts for Redraw

Our method utilizes a composite prompt $\text{pt}_0 + \sum_{i=1} \text{pt}_i$ to facilitate the redrawing of erased regions while promoting result diversity. In this formulation, $\text{pt}_0$ enforces the constraint to preserve the context in the unmasked region $(1 - M)$, while each $\text{pt}_i$ represents an objective to diverge from the context of the $i$-th generation within the masked region $M$. Guiding the generation with these prompts is equivalent to sampling from a reshaped distribution $\hat{p}_n(\boldsymbol{x})$ as defined in Equation (5), where the energy function models the conditional probability of the composite prompt $\text{pt}_0 + \sum_{i=1} \text{pt}_i$.

We record the intermediate states of each generation to facilitate diversity sampling. Let $\boldsymbol{x}_{t_n}^{(i)}$ denote the $i$-th generation results at time step $t_n$. For the $(i+1)$-th iteration, the newly generated sample must maintain consistency within the unmasked region $(1 - M)$, while the content within region $M$ must deviate from the set of previous results $\{\boldsymbol{x}_{t_n}^{(j)}\}_{j=1}^i$ to promote diversity. To formalize these constraints, we define an energy function $E_n(\boldsymbol{x})$ that assigns lower energy to states $\boldsymbol{x}$ that satisfy these requirements:

$$E_n(\boldsymbol{x}) = \lambda \underbrace{\|(1 - M) \odot (\boldsymbol{x} - \boldsymbol{x}_{t_n}^{(1)})\|_2^2}_{\text{Context Preservation for Unmask Region}}$$
$$- \zeta \underbrace{\sum_{j=1}^i \|M \odot (\boldsymbol{x} - \boldsymbol{x}_{t_n}^{(j)})\|_2^2}_{\text{Diversity Repulsion for Mask Region}} \tag{7}$$

where the first term enforces context preservation in the unmasked areas, and the second term acts as a repulsive force to drive the new generation away from prior samples in the masked region.

We incorporate this energy gradient into Equation (21) to construct a steered velocity field $\boldsymbol{v}_n^f$. According to Theorem A.2, this modified dynamics ensures that the sampled trajectories evolve toward the reshaped distribution $\hat{p}_n(\boldsymbol{x})$.

$$\hat{\boldsymbol{v}}_n^f(\boldsymbol{x}, \text{pt}) = \boldsymbol{v}_n(\boldsymbol{x}, \text{pt}) + \eta \nabla_{\boldsymbol{x}} E_n(\boldsymbol{x}) \tag{8}$$

### 3.3. Stochastic Prompts for Erasure

The redrawing process follows a deterministic straight path, making the output highly sensitive to the initial seed and textual prompt pt. As illustrated in Figure 1, a straight-path ODE often converge to a suboptimal mode, yielding only a gray and human-like robot, despite different random seeds. Thus, enhancing diversity is critical for artistic expressiveness, as a broader set of samples increases the likelihood of satisfying complex user constraints without semantic drift (Xie et al., 2025; Zhuo et al., 2025).

To facilitate mode-switching without altering the noise and prompt, we introduce a *zigzag (backward-and-forward)* refinement path. Specifically, given an intermediate state $\boldsymbol{x}_{t_{n+d}}$, we employ the reverse mapping to compute an inverted state $\tilde{\boldsymbol{x}}_{t_n}$, which then serves as a novel initialization for subsequent forward process (2). We refer to this as the *Erasure* process, as it dissolves the details of the original state to catalyze mode-switching, allowing the trajectory to escape local mode entrapment and explore alternative semantic regions.

The necessity of this stochasticity arises from the invertibility of ODE systems; utilizing the same velocity field for the reverse process would yield an inverted state $\tilde{\boldsymbol{x}}_{t_n}$ identical to the original $\boldsymbol{x}_{t_n}$. To mitigate this determinism, we introduce stochastic prompts during the backward phase. We construct a stochastic prompt ps by augmenting pt with random words to perturb the semantic guidance

$$\text{ps}_n = \text{pt} + \text{random words} \tag{9}$$

The reverse mapping steered by $\boldsymbol{v}_n^b$ ensures $\tilde{\boldsymbol{x}}_{t_n}$ deviates from $\boldsymbol{x}_{t_n}$, enabling the exploration of new modes:

$$\hat{\boldsymbol{v}}_n^b(\boldsymbol{x}, \text{ps}_n, \text{pt}) = (1+w)\hat{\boldsymbol{v}}_n^f(\boldsymbol{x}, \text{ps}_n) - w\hat{\boldsymbol{v}}_n^f(\boldsymbol{x}, \text{pt})$$
$$= (1+w)\boldsymbol{v}_n(\boldsymbol{x}, \text{ps}_n) - w\boldsymbol{v}_n(\boldsymbol{x}, \text{pt}) + \eta\nabla_{\boldsymbol{x}}E_n(\boldsymbol{x}), \tag{10}$$

where $w$ steers the inverted state toward novel semantic regions. The formulation $(1+w)\boldsymbol{v}_n(\boldsymbol{x}, \text{ps}_n) - w\boldsymbol{v}_n(\boldsymbol{x}, \text{pt})$ amplifies the stochastic direction introduced by $\text{ps}_n$ while explicitly deviating from the original trajectory $\boldsymbol{v}_n(\boldsymbol{x}, \text{pt})$ computed from pt. Finally, the term $\eta\nabla_{\boldsymbol{x}}E_n(\boldsymbol{x})$ constrains $\tilde{\boldsymbol{x}}_{t_n}$ to maintain global context in unmasked regions while encouraging divergence in masked regions. This ensures $\tilde{\boldsymbol{x}}_{t_n}$ achieves a controlled stochastic deviation from the original state without losing structural coherence.

The optimal timing for erasure is determined by the alignment to the visual prompt $\text{p}_0 + \sum_{i=1}\text{p}_i$. Intuitively, we expect the guided transition to satisfy $\hat{p}_n(\boldsymbol{x}_{t_{n+1}}|\boldsymbol{x}_{t_n}) > p_n(\boldsymbol{x}_{t_{n+1}}|\boldsymbol{x}_{t_n})$. According to Theorem A.2, this implies that the energy should decrease as the state aligns with the visual prompts, *i.e.* $E(\boldsymbol{x}_{t_{n+1}}) < E(\boldsymbol{x}_{t_n})$. If the result fails to decrease $E(\boldsymbol{x}_{t_{n+1}}) \geq E(\boldsymbol{x}_{t_n})$, we conclude the current step has not effectively integrated the guidance. In such instances, we trigger the zigzag refinement process.

---

**Algorithm 1** Erasure-Redraw Sampling

1: **Input:** number $N$ for total timesteps, number $\delta$ for erasure steps, textual prompt pt.
2: $\boldsymbol{x}_{t_0} \sim p_0(\boldsymbol{x})$
3: **for** $n = 0, \dots, N$ **do**
4:    $\boldsymbol{x}_{t_{n+1}} \leftarrow \boldsymbol{x}_{t_n} + \hat{\boldsymbol{v}}_n(\boldsymbol{x}_{t_n}, \text{pt})\Delta t$
5:    **if** $E(\boldsymbol{x}_{t_{n+1}}) \geq E(\boldsymbol{x}_{t_n})$ **then**
6:       **for** $k = n+1, \dots, n+1-\delta$ **do**
7:          Sample Stochastic Prompt $\text{ps}_k$ via Equation (9).
8:          Compute $\hat{\boldsymbol{v}}_k^b(\boldsymbol{x}_{t_k}, \text{ps}_k, \text{pt})$ via Equation (10)
9:          $\boldsymbol{x}_{t_{k-1}} \leftarrow \boldsymbol{x}_{t_k} - \hat{\boldsymbol{v}}_k^b(\boldsymbol{x}_{t_k}, \text{ps}_k, \text{pt})\Delta t$
10:       **end for**
11:       **for** $j = n+1-\delta, \dots, , n$ **do**
12:          Compute $\hat{\boldsymbol{v}}_k^f(\boldsymbol{x}_{t_k}, \text{pt})$ via Equation (8)
13:          $\boldsymbol{x}_{t_{j+1}} \leftarrow \boldsymbol{x}_{t_j} + \hat{\boldsymbol{v}}_j^f(\boldsymbol{x}_{t_j}, \text{pt})\Delta t$
14:       **end for**
15:    **end if**
16: **end for**
17: $\boldsymbol{x}_1 \leftarrow \boldsymbol{x}_{t_N}$
18: **return** $\boldsymbol{x}_1$

---

### 3.4. Algorithm for Erasure-Redraw Sampling

We present the Erasure-Redraw Sampling procedure in Algorithm 1. The process begins with a forward denoising step in line 4. Subsequently, the algorithm determines whether to perform zigzag refinement to introduce additional semantic deviation. If required, the backward and forward processes are executed in lines 9 and 13, respectively.

**Theorem 3.3** (Proof is left in the supplementary material)**.**
*If we first denoise $\boldsymbol{x}_{t_{n+1}}$ to obtain $\boldsymbol{x}_{t_n}$ and then we invert $\tilde{\boldsymbol{x}}_{t_n} = \boldsymbol{x}_{t_n}$ to get $\tilde{\boldsymbol{x}}_{t_{n+1}}$ for each timestep. The cumulative semantic information difference $\delta = \sum_n(\hat{\boldsymbol{x}}_{t_n} - \boldsymbol{x}_{t_n})^2$ can be written as*

$$\delta = (\Delta t)^2 \sum_n \Big( \underbrace{\hat{\boldsymbol{v}}_n^f(\boldsymbol{x}_{t_n}, \text{pt}) - \hat{\boldsymbol{v}}_n^b(\boldsymbol{x}_{t_n}, \text{ps}_n, \text{pt})}_{\text{Semantic Drift}} +$$
$$\underbrace{\hat{\boldsymbol{v}}_n^b(\boldsymbol{x}_{t_n}, \text{ps}_n, \text{pt}) - \hat{\boldsymbol{v}}_{n+1}^b(\boldsymbol{x}_{t_{n+1}}, \text{ps}_{n+1}, \text{pt})}_{\text{Approximation Error}} \Big)^2. \tag{11}$$

Our Erasure-Redraw Sampling algorithm is specifically designed to introduce semantic deviation to facilitate diverse generation. At each zigzag (erasure-redraw) step, we can quantify this deviation by $(\hat{\boldsymbol{x}}_{t_n} - \boldsymbol{x}_{t_n})^2$; the cumulative semantic deviation thus serves as a metric for the total information injected into the generation process. While a general formulation for Erasure-Redraw Sampling is complex, Theorem 3.3 characterizes a simplified case. The theorem decomposes semantic deviation into *Semantic Drift* and *Approximation Error*. This reveals that the semantic shifts, guided by stochastic and visual prompts, are the fundamental contributors to the model's creative output.

*"a crystal ball with a flower in the rocks and mud"*

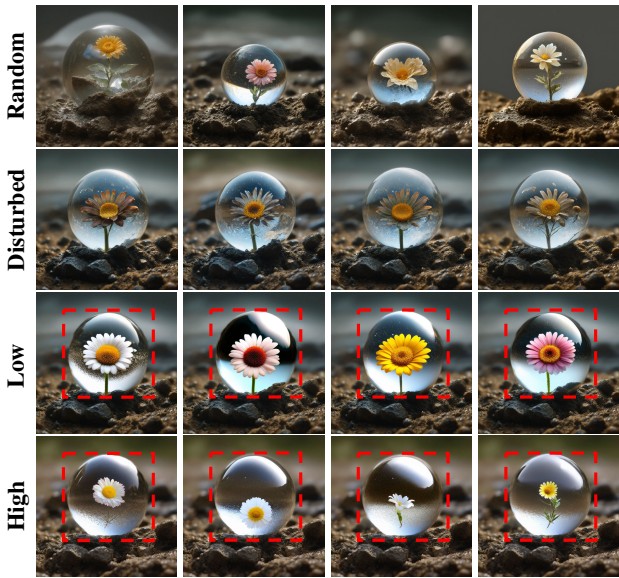

*Figure 2.* Controllability of Diversity Strength. Row 1: Baseline I.I.D. sampling using distinct noise seeds. Row 2: Results from a single noise with additive perturbations. Rows 3–4: Our method's multi-granularity diversity control via zigzag path. Specifically, Row 3 demonstrates low-strength diversity (color and texture variations) achieved with $\delta = 2$ backward steps, whereas Row 4 illustrates high-strength diversity (structural and geometric reconfiguration) using $\delta = 8$ backward steps. Compared to the stochastic baselines (Rows 1–2), our method enables targeted semantic deviations that are unattainable through simple initialization heuristics. Red boundaries indicate the masks used for localized refinement.

## 4. Experiments

In this section, we conduct an extensive evaluation to demonstrate the capabilities and properties of our method.

### 4.1. Experimental Setup

**Generative Models** Diversity research in diffusion and flow matching models remains sparse compared to GANs (Liu et al., 2020; Yildirim et al., 2023). Existing work primarily addresses diffusion paradigms (Kumari et al., 2023; Miao et al., 2024; Corso et al., 2024; Sadat et al., 2024), leaving the diversity of flow matching models largely underexplored. We conduct a comparison of several state-of-the-art flow matching methods against the standard baseline of sampling independent and identically distributed (I.I.D.) noise. Comparison methods are categorized into: (1) publicly available frameworks, including PromptMoG (Ruan et al., 2025), SPARKE (Jalali et al., 2025), and Group (Parmar et al., 2026), discussed in the main text; and (2) proprietary or unavailable methods, specifically SPELL (Kirchhof et al., 2025), DiverseFlow (Morshed & Boddeti, 2025), and CNO (Kim et al., 2026), for which our re-implementations are detailed in the supplementary material.

**Datasets and Metrics** Our approach seeks to push the boundaries of the quality-diversity trade-off without compromising semantic alignment with the input text. To quantitatively evaluate our approach, we employ a comprehensive suite of metrics: (i) Consistency : we use CLIPScore (Hessel et al., 2021) to measure text–image consistency. (ii) Quality: we use Aes Score (Schuhmann et al., 2022) and PickScore (Kirstain et al., 2023) to assess semantic relevance and aesthetic appeal; (iii) Diversity: we adopt the Mean Similarity Score (MSS) (Sadat et al., 2024), Vendi Score (Friedman & Dieng, 2023) and In-Batch Similarity Score (IBS Score) (Corso et al., 2024) to quantify the variation among generated samples. All experiments are conducted using text prompts randomly sampled from the GenEval (Ghosh et al., 2023) and MS-COCO (Lin et al., 2014) validation sets. For each prompt, we generate 3–5 images to facilitate a robust comparative analysis.

### 4.2. Multi-granularity Diversity

Our framework enables precise control over diversity across both spatial and semantic dimensions. By modulating the mask $M$ within the energy function $E_n(\mathbf{x})$, we achieve spatial granularity control, facilitating both localized refinement and global structural exploration. Unlike existing methods that rely on coarse, unconstrained stochasticity, our zigzag path introduces intensity-based control via the backward step size $\delta$. This temporal manipulation allows for a hierarchical exploration of the data manifold: Low-intensity variation: Minimal backward intervals (erasing little) preserve the original geometric topology while varying color (Fig. 2, Row 3). High-intensity variation: Larger backward steps (erasing more) enable the trajectory to escape local modes, facilitating structural reconfiguration and geometric shifts (Fig. 2, Row 4). This fine-grained control represents a significant departure from traditional noise-injection strategies, which lack structural and intensity constraints. As shown in Fig. 2, while I.I.D. noise (Row 1) and additive perturbations (Row 2) yield unpredictable variations, our method consistently achieves targeted semantic deviations that are unattainable through simple initialization heuristics.

### 4.3. Qualitative and Quantitative Comparison

**Qualitative Comparison** We evaluate whether incorporating visual and stochastic prompts into the zigzag path enhances the diversity of flow matching models. Following Section 4.1, Figure 7 (rows 1–4) displays results from I.I.D. sampling, PromptMoG (Ruan et al., 2025), SPARKE (Jalali et al., 2025), and Group (Parmar et al., 2026). In contrast, our method introduces a masking mechanism that decouples context preservation from diversity injection, enabling foreground modification while keeping the background static. The degree of semantic deviation is controlled by the number $\delta$ of erasure steps in the zigzag path: $\delta = 2$ primarily

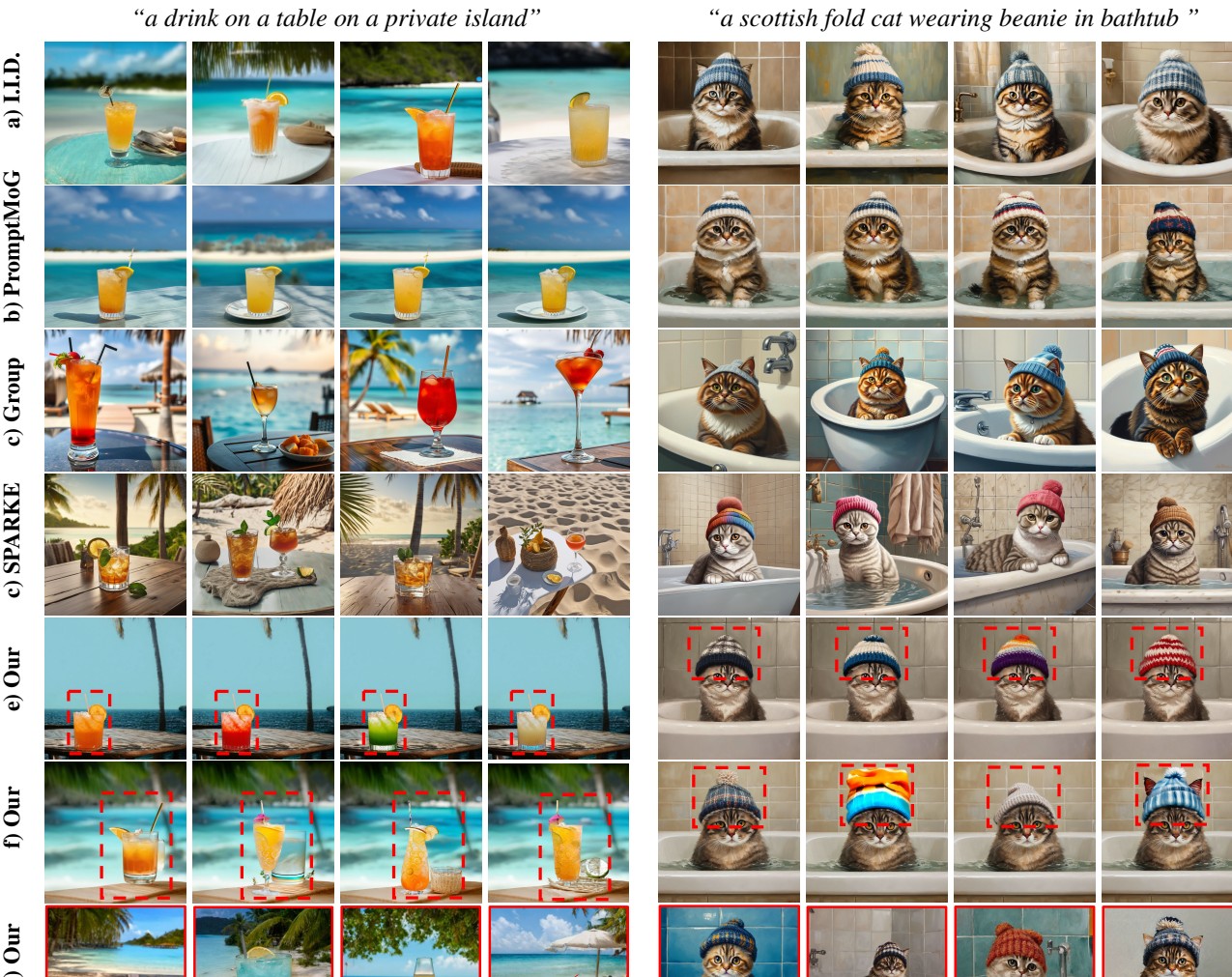

*Figure 3.* Qualitative Comparison for Diversity Generation. The figure evaluates diversity across two prompts: "A drink on a table on a private island" (left) and "A Scottish Fold cat wearing a beanie in a bathtub" (right). The top four rows display results from baseline methods: I.I.D. sampling, PromptMoG, Group, and SPARKE. The final three rows demonstrate our method's capacity for multi-level variations: (i) color variations, (ii) shape/geometry variations, and (iii) scenery/contextual variations.

induces color variations (row 5), whereas deeper erasure ($\delta = 8$) removes structural details, facilitating significant shape and geometric variations (row 6). When the mask covers the entire image, our approach effectively generates a completely new sample (row 7), aligning with the global diversity achieved by baseline methods.

**Quantitative Comparison**  In Table 1, we evaluate our method against baselines across two backbones, including SD3.5 (Esser et al., 2024), and FLUX.1 (Black Forest Labs, 2024). The results underscore three key observations: First, our method consistently outperforms baselines in diversity metrics. This indicates that our sampling effectively explores broader modes of the data distribution. Second, while

existing diversity-enhancing methods often suffer from performance degradation in terms of semantic alignment, our approach preserves high-fidelity results. Finally, the consistent gains across diverse backbones demonstrate the generalizability of our approach.

**Efficiency Comparison**  In text-to-image systems, independent sampling often yields redundant results due to mode collapse. While group inference improves mode coverage by generating samples simultaneously, it linearly scales GPU memory consumption with the number of samples, confining such methods to enterprise-grade hardware. To resolve this, we propose a memory-efficient sequential sampling framework. By introducing visual prompts, we enforce the

*Table 1.* Quantitative Comparison for Diversity Generation. Experimental results demonstrate that our proposed method consistently outperforms alternative inference strategies. Across three distinct architectures, our approach yields systematic improvements in the generative diversity of text-to-image flow matching models. The best-performing results are highlighted in bold.

| Backbone | Methods | Consistency | Quality | | Diversity | | |
| --- | --- | --- | --- | --- | --- | --- | --- |
| | | CLIP Score↑ | Aes Score↑ | PickScore↑ | MSS↓ | Vendi Score↑ | IBS Score↓ |
| SD3.5 | I.I.D. | 0.2193 | 5.5742 | 22.546 | 0.1984 | 3.5931 | 0.7242 |
| | PromptMoG | 0.2097 | 5.5672 | 22.705 | 0.1840 | 3.7491 | 0.7029 |
| | Group | 0.2204 | **5.7031** | 22.941 | 0.1297 | 3.9410 | 0.6631 |
| | SPARKE | 0.2183 | 5.3675 | 22.864 | **0.1204** | 3.8509 | 0.6765 |
| | Our | **0.2234** | 5.5917 | **23.013** | 0.1305 | **3.9513** | **0.6328** |
| FLUX.1 | I.I.D. | 0.2201 | 5.8531 | 22.806 | 0.1749 | 3.7385 | 0.6947 |
| | PromptMoG | 0.2105 | **5.9463** | 22.893 | 0.1582 | 3.9614 | 0.6828 |
| | Group | 0.2231 | 5.9295 | **22.913** | **0.1243** | 4.0681 | 0.6242 |
| | SPARKE | 0.2215 | 5.6931 | 22.874 | 0.1459 | 3.9571 | 0.6489 |
| | Our | **0.2234** | 5.9184 | 22.906 | 0.1271 | **4.1614** | **0.6096** |

*Table 2.* Efficiency Comparison for Diversity Generation. Unlike joint sampling methods, our approach reformulates parallel inference into a sequential paradigm using visual prompts. This iterative processing ensures a constant memory footprint, enabling diverse generation on resource-constrained consumer GPUs.

| Method | VRAM (GB) | Method | VRAM (GB) |
| --- | --- | --- | --- |
| Group | 36.89 | DiverseFlow | 32.63 |
| SPELL | 26.43 | Our | 17.87 |

$i$-th sample to deviate from previously generated results $\{\boldsymbol{x}^{(j)}\}_{j=1}^{i-1}$ via the repulsive term $\sum_{j=1}^{i-1}\|M \odot (\boldsymbol{x} - \boldsymbol{x}_{t_n}^{(j)})\|_2^2$. This shift to a sequential paradigm ensures a constant memory footprint regardless of the total sample count, as only one image is processed at any time. As shown in Table 2, this "time-for-memory" trade-off democratizes high-diversity generation for consumer-grade GPUs.

### 4.4. Applications

**Image Protection**  We evaluate our method's capacity to mitigate training data memorization. We trained a Flow Matching model (Lipman et al., 2024) on *ImageNet* (Deng et al., 2009), which tends to replicate training instances as shown in Figure 4. By injecting visual prompts, our strategy encourages the model to bypass high-probability training modes and explore novel regions of the data manifold.

While conceptually similar to Sparse Repellency in Diverse-Flow (Morshed & Boddeti, 2025), our approach differs by computing repellency directly within the latent trajectory, rather than mapping intermediate states back to the clean image space. Furthermore, according to Theorem 3.3, our stochastic prompts provide an additional deviation mechanism, ensuring generated samples remain distinct from protected references.

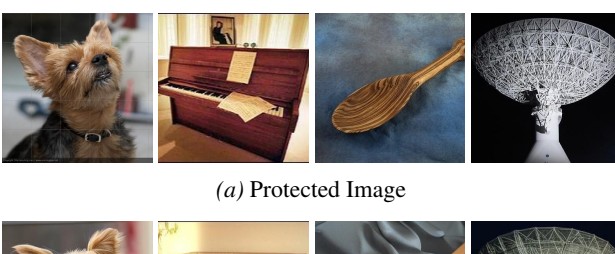

*(a)* Protected Image

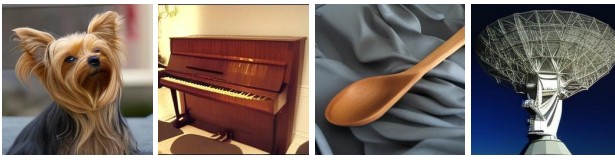

*(b)* SD 3.5

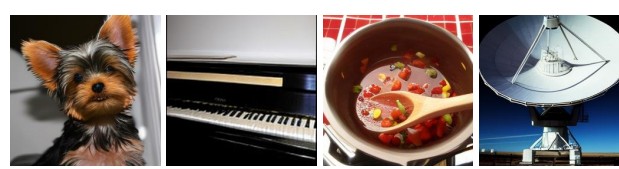

*(c)* SD 3.5 + Our Sampling Method

*Figure 4.* Image Protection. The second row illustrates the standard model's tendency to replicate training instances shown in the first row. In contrast, when integrated with our method and initiated from the same noise and prompts, the model generates distinct and novel samples (third row) that deviate from the training data.

**Image Recommendation**  While flow matching models excel at instruction following, they seldom yield optimal results in a single inference pass. Typically, users favor specific components of a generation while requiring diverse variations in other regions to iteratively refine the output. Increased diversity broadens the search space, highering the probability of satisfying complex user preferences. Our approach facilitates this by providing a diverse array of candidates, thereby enhancing the efficiency of the human-in-the-loop recommendation process, as illustrated in Figure 5.

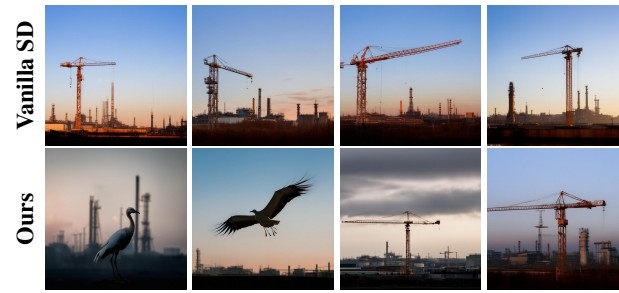

*Figure 5.* Image Recommendation. The top row displays the initial generation, the binary mask $M$ defining the target region, and the corresponding textual prompt. The second row showcases semantically diverse variations generated within the masked area. By exploring alternative visual modes while preserving global context, our method facilitates the refinement of unsatisfactory local details without re-sampling the entire image.

*Table 3.* Impact of Stochastic Prompt Composition. We evaluate the sensitivity of our method to the source of stochasticity by composing $\text{ps}_n$ from: (i) random prompts sampled from the MS-COCO dataset, and (ii) arbitrary words selected from a general corpus. The consistent diversity observed across both strategies demonstrates that the structural perturbation to the velocity field is robust to the specific semantic source of the stochastic prompt.

| Method | MSS | Vendi Score | IBS Score |
|--------|-----|-------------|-----------|
| Dataset | 0.1305 | 3.9513 | 0.6328 |
| Corpus | 0.1287 | 3.9581 | 0.6325 |

**Polysemous Prompts** A prompt's conditional distribution often spans multiple semantic modes. Our objective is to capture these variations in an efficient manner. As shown in Figure 1, a user's specific intent—often defined by particular colors or shapes—may mismatch the model's initial deterministic output. By generating diverse candidates, our method reduces the sampling attempts. This is vital for polysemous prompts with ambiguous interpretations. Figure 6 demonstrates that our method successfully captures the dual meanings of "crane"—mechanical equipment and biological entities—thereby providing comprehensive coverage of the underlying data distribution.

### 4.5. Ablation Study

Existing diversity-enhancing methods typically inject randomness into the forward process. In contrast, our method utilizes stochastic prompts $\text{ps}_n$ within an erasure-redraw path to introduce stochasticity during the backward process, preserving the initial noise and prompt pt. As defined in Equation (9), this modifies the backward velocity field $\hat{v}_n^b(\boldsymbol{x}, \text{ps}_n, \text{pt})$, enabling trajectory exploration distinct from the prompt-guided direction $\boldsymbol{v}_n(\boldsymbol{x}, \text{pt})$ and the repulsive gradient $\nabla_{\boldsymbol{x}} E_n(\boldsymbol{x})$. We investigate the impact of stochastic word selection and the hyper-parameters $\{\lambda, \zeta\}$ in $E_n$.

*"**crane** against industrial city at the crack of dawn"*

Vanilla SD

Ours

*Figure 6.* Polysemous Prompts. The text prompt (top) contains an underscored polysemous word. While the default results from Stable Diffusion (first row) suffer from semantic collapse—capturing only a single interpretation—our method (second row) successfully recovers the multi-modal nature of the prompt. By exploring the data manifold more effectively, our approach generates diverse instances corresponding to distinct semantic meanings of the same word, demonstrating superior coverage of the prompt's underlying distribution.

We evaluate two strategies for composing the stochastic prompt $\text{ps}_n$ in Equation (9): (i) selecting random prompts from the MS-COCO dataset, and (ii) selecting arbitrary words from a general corpus. The resulting diversity scores are listed in Table 3, demonstrating that our method is robust to the specific semantic content of the stochastic prompts.

By setting $\lambda = \zeta$, we sweep these parameters across the range $\{0.05, 0.1, 0.5, 1.0, 2.0\}$. The resulting MSS scores—$\{0.1291, 0.1306, 0.1305, 0.1308, 0.1301\}$, respectively—remain remarkably stable. Unlike the Sparse Repellency in DiverseFlow (Morshed & Boddeti, 2025), where small coefficients lead to negligible deviation, our stochastic prompts maintain diversity even when repulsive forces are weak. This confirms that the stochastic velocity field provides a secondary, robust mechanism for mode exploration.

For all experiments, we maintain a guidance scale of $w = 5.5$ and $\eta = 1$. The influence of the guidance scale is well-documented in the literature (Ho & Salimans, 2021).

### 5. Conclusion

In this paper, we present *Erasure-Redraw Sampling*, a framework that introduces a *zigzag path* into the standard straight-path ODE sampling process to enhance generative diversity. While flow-based models typically define a deterministic mapping from source to sample, our method modifies this mapping at inference time by injecting stochasticity into the deterministic process via zigzag path. This allows for the exploration of diverse semantic modes that are often bypassed by standard solvers. We demonstrate the utility of our approach across several applications and provide extensive experimental evidence showcasing its advantages in both diversity and sample efficiency.

## Impact Statement

This paper presents work whose goal is to advance the field of Computer Vision. There are many potential societal consequences of our work, none which we feel must be specifically highlighted here.

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

## A. Theorem Proof

**Theorem A.1.** *The transition intensity of the reshaped distribution $\hat{p}_n$ can be expressed as a reweighting of the original transition intensity by the exponential change in energy:*

$$\hat{p}_n(\boldsymbol{x}_{t_{n+1}}|\boldsymbol{x}_{t_n}) = p_n(\boldsymbol{x}_{t_{n+1}}|\boldsymbol{x}_{t_n})\exp(E_n(\boldsymbol{x}_{t_{n+1}}) - E_n(\boldsymbol{x}_{t_n})) \tag{12}$$

*Proof.* Consider the reshaped marginal distribution at a discrete step $t_n$, modulated by the energy function $E(\mathbf{x})$:

$$\hat{p}_n(\mathbf{x}_{t_n}) = \frac{1}{Z_n}p_n(\mathbf{x}_{t_n})\exp(E_n(\mathbf{x}_{t_n})), \tag{13}$$

where $Z_n$ is the partition function and $E_n(\boldsymbol{x})$ is defined to balance context fidelity and diversity:

$$E_n(\boldsymbol{x}) = \lambda\|(1-M)\odot(\boldsymbol{x}-\boldsymbol{x}_{t_n}^{(1)})\|_2^2 - \eta\sum_{i=1}^{l}\|M\odot(\boldsymbol{x}-\boldsymbol{x}_{t_n}^{(i)})\|_2^2. \tag{14}$$

To derive the transition intensity $\hat{p}_n(\mathbf{x}_{t_{n+1}}|\mathbf{x}_{t_n})$, we first define the joint distribution $\hat{p}_n(\mathbf{x}_{t_{n+1}}, \mathbf{x}_{t_n})$ by applying the energy reshaping to the terminal state of the transition:

$$\hat{p}_n(\mathbf{x}_{t_{n+1}}, \mathbf{x}_{t_n}) = \frac{1}{Z_{n+1}}p_n(\mathbf{x}_{t_{n+1}}, \mathbf{x}_{t_n})\exp(E_{n+1}(\mathbf{x}_{t_{n+1}})) \tag{15}$$

$$= \frac{1}{Z_{n+1}}p_n(\mathbf{x}_{t_{n+1}}|\mathbf{x}_{t_n})p_n(\mathbf{x}_{t_n})\exp(E_{n+1}(\mathbf{x}_{t_{n+1}})). \tag{16}$$

By the definition of conditional probability, the reshaped transition intensity is:

$$\hat{p}_n(\mathbf{x}_{t_{n+1}}|\mathbf{x}_{t_n}) = \frac{\hat{p}_n(\mathbf{x}_{t_{n+1}}, \mathbf{x}_{t_n})}{\hat{p}_n(\mathbf{x}_{t_n})} \tag{17}$$

$$= \frac{\frac{1}{Z_{t_{n+1}}}p_n(\mathbf{x}_{t_{n+1}}|\mathbf{x}_{t_n})p_n(\mathbf{x}_{t_n})\exp(E_{t_{n+1}}(\mathbf{x}_{t_{n+1}}))}{\frac{1}{Z_{t_n}}p_n(\mathbf{x}_{t_n})\exp(E(\mathbf{x}_{t_n}))} \tag{18}$$

$$= p_n(\mathbf{x}_{t_{n+1}}|\mathbf{x}_{t_n})\frac{Z_{t_n}}{Z_{t_{n+1}}}\frac{\exp(E(\mathbf{x}_{t_{n+1}}))}{\exp(E(\mathbf{x}_{t_n}))}. \tag{19}$$

Assuming that within an infinitesimal time step $\Delta t = t_{n+1} - t_n$, the partition function $Z$ remains constant ($Z_{n+1} \approx Z_n$) and the energy function parameters $\lambda, \eta$ are locally stationary, the transition simplifies to:

$$\hat{p}_n(\boldsymbol{x}_{t_{n+1}}|\boldsymbol{x}_{t_n}) = p_n(\boldsymbol{x}_{t_{n+1}}|\boldsymbol{x}_{t_n})\exp\left(E_n(\boldsymbol{x}_{t_{n+1}}) - E_n(\boldsymbol{x}_{t_n})\right). \tag{20}$$

Substituting the definition of $E_n(\boldsymbol{x})$, the exponential term acts as a steering factor that increases the probability of transitions toward states $\boldsymbol{x}_{t_{n+1}}$ that maintain context (minimizing the first term) and maximize diversity (maximizing the second term relative to $E_n(\boldsymbol{x}_{t_n})$). $\square$

**Theorem A.2.** *The velocity field $\hat{v}_n^f(\boldsymbol{x}_{t_n}, \mathrm{pt})$ that yields the reshaped transition $\hat{p}(\boldsymbol{x}_{t_{n+1}}|\boldsymbol{x}_{t_n}) = p(\boldsymbol{x}_{t_{n+1}}|\boldsymbol{x}_{t_n})\exp(E_n(\boldsymbol{x}_{t_{n+1}}) - E_n(\boldsymbol{x}_{t_n}))$ is established by the following theorem.*

$$\hat{v}_n^f(\boldsymbol{x}_{t_n}, \mathrm{pt}) = v_n^f(\boldsymbol{x}_{t_n}, \mathrm{pt}) + \eta\nabla_{\boldsymbol{x}}E_n(\boldsymbol{x}_{t_n}) + \mathcal{O}(\Delta t), \tag{21}$$

*Proof.* We aim to derive the guided velocity field $\hat{\mathbf{v}}_t(\mathbf{x})$ that transforms a baseline velocity field $\mathbf{v}_t(\mathbf{x})$ to satisfy the reshaped distribution $\hat{p}_t(\mathbf{x}) = \frac{1}{Z_t}p_t(\mathbf{x})\exp(E_t(\mathbf{x}))$. We provide a dual-perspective derivation: one from the continuous continuity equation and another from the discrete trajectory perturbation.

The evolution of the baseline density $p_t(\mathbf{x})$ is governed by the continuity equation:

$$\frac{\partial p_t(\mathbf{x})}{\partial t} = -\nabla\cdot(p_t(\mathbf{x})\mathbf{v}_t(\mathbf{x})). \tag{22}$$

Taking the gradient of the log-density for the reshaped distribution yields the modified score function:

$$\nabla_{\mathbf{x}} \log \hat{p}_t(\mathbf{x}) = \nabla_{\mathbf{x}} \log p_t(\mathbf{x}) + \nabla_{\mathbf{x}} E_t(\mathbf{x}). \tag{23}$$

In the context of probability flow ODEs, the velocity field $\mathbf{v}_t$ is inherently coupled to the score function. Specifically, any modification to the density's log-gradient necessitates a corresponding correction $\mathbf{u}_t(\mathbf{x})$ to the velocity field, such that $\hat{\mathbf{v}}_t = \mathbf{v}_t + \mathbf{u}_t$. Substituting Eq. 23 into the Fokker-Planck or Continuity framework implies that the corrective drift must align with the energy gradient to maintain the reshaped density flow.

To bridge this with numerical inference, we utilize the result from Theorem 3.1, where the reshaped transition intensity is:

$$\hat{p}_n(\mathbf{x}_{t_{n+1}}|\mathbf{x}_{t_n}) = p_n(\mathbf{x}_{t_{n+1}}|\mathbf{x}_{t_n}) \exp(E_n(\mathbf{x}_{t_{n+1}}) - E_n(\mathbf{x}_{t_n})). \tag{24}$$

For a small discretization step $\Delta t$, we Taylor expand the energy function $E_n(\mathbf{x}_{t_{n+1}})$ around $\mathbf{x}_{t_n}$ along the trajectory $\Delta\mathbf{x} = \mathbf{x}_{t_{n+1}} - \mathbf{x}_{t_n}$:

$$\Delta E = E_n(\mathbf{x}_{t_{n+1}}) - E_n(\mathbf{x}_{t_n}) = \nabla_{\mathbf{x}} E_n(\mathbf{x}_{t_n})^\top (\mathbf{x}_{t_{n+1}} - \mathbf{x}_{t_n}) + \mathcal{O}(\Delta t). \tag{25}$$

Substituting the discrete mapping $\mathbf{x}_{t_{n+1}} - \mathbf{x}_{t_n} = \boldsymbol{v}_n^f(\boldsymbol{x}_{t_n}, \mathrm{pt})\Delta t$ into the exponent, the energy gain $\Delta E$ acts as a local reweighting of the transition. To realize this reweighting as a displacement in the velocity field, we define the guided velocity field $\hat{\mathbf{v}}_n^f$ as:

$$\hat{\boldsymbol{v}}_n^f(\boldsymbol{x}_{t_n}, \mathrm{pt}) = \boldsymbol{v}_n^f(\boldsymbol{x}_{t_n}, \mathrm{pt}) + \eta \nabla_{\boldsymbol{x}} E_n(\boldsymbol{x}_{t_n}) + \mathcal{O}(\Delta t), \tag{26}$$

where $\eta$ is the guidance scale (acting as an inverse temperature). This additive correction ensures that each discrete step moves the sample towards higher-energy regions defined by $E(\mathbf{x})$, while remaining consistent with the underlying probability flow ODE.

The two perspectives are equivalent as $\Delta t \to 0$, confirming that gradient guidance is the infinitesimal realization of the energy-based distribution reshaping. $\qquad\square$

**Theorem A.3.** *If we first denoise $\boldsymbol{x}_{t_{n+1}}$ to obtain $\boldsymbol{x}_{t_n}$ and then we invert $\tilde{\boldsymbol{x}}_{t_n} = \boldsymbol{x}_{t_n}$ to get $\tilde{\boldsymbol{x}}_{t_{n+1}}$ for each timestep. The cumulative semantic information difference $\delta = \sum_n (\hat{\boldsymbol{x}}_{t_n} - \boldsymbol{x}_{t_n})^2$ can be written as*

$$\delta = (\Delta t)^2 \sum_n \Big( \underbrace{\hat{\boldsymbol{v}}_n^f(\boldsymbol{x}_{t_n}, \mathrm{pt}) - \hat{\boldsymbol{v}}_n^b(\boldsymbol{x}_{t_n}, \mathrm{ps}_n, \mathrm{pt})}_{\text{Semantic Drift}} +$$
$$\underbrace{\hat{\boldsymbol{v}}_n^b(\boldsymbol{x}_{t_n}, \mathrm{ps}_n, \mathrm{pt}) - \hat{\boldsymbol{v}}_{n+1}^b(\boldsymbol{x}_{t_{n+1}}, \mathrm{ps}_{n+1}, \mathrm{pt})}_{\text{Numerical Error}} \Big)^2. \tag{27}$$

*Proof.* Consider a single discrete step. We start with the state $\mathbf{x}_{t_{n+1}}$ at time $t_{n+1}$. First, we perform the backward (denoising) step to find the state at $t_n$:

$$\mathbf{x}_{t_n} = \mathbf{x}_{t_{n+1}} - \hat{\mathbf{v}}_{n+1}^b(\mathbf{x}_{t_{n+1}}, \mathrm{ps}_{n+1}, \mathrm{pt})\Delta t \tag{28}$$

Next, we apply the forward (inversion) step starting from $\mathbf{x}_{t_n}$ using the reshaped velocity field $\hat{\mathbf{v}}_n^f$:

$$\tilde{\mathbf{x}}_{t_{n+1}} = \mathbf{x}_{t_n} + \hat{\mathbf{v}}_n^f(\mathbf{x}_{t_n}, \mathrm{pt})\Delta t \tag{29}$$

Substituting the expression for $\mathbf{x}_{t_n}$ into the forward step:

$$\tilde{\mathbf{x}}_{t_{n+1}} = \big(\mathbf{x}_{t_{n+1}} - \hat{\mathbf{v}}_{n+1}^b(\mathbf{x}_{t_{n+1}}, \mathrm{ps}_{n+1}, \mathrm{pt})\Delta t\big) + \hat{\mathbf{v}}_n^f(\mathbf{x}_{t_n}, \mathrm{pt})\Delta t \tag{30}$$

Rearranging the terms to find the local difference $\Delta\mathbf{x}_{n+1} = \tilde{\mathbf{x}}_{t_{n+1}} - \mathbf{x}_{t_{n+1}}$:

$$\Delta\mathbf{x}_{n+1} = \big([\hat{\mathbf{v}}_n^f(\mathbf{x}_{t_n}, \mathrm{pt}) - \hat{\mathbf{v}}_{n+1}^b(\mathbf{x}_{t_{n+1}}, \mathrm{ps}_{n+1}, \mathrm{pt})]\Delta t\big)^2 \tag{31}$$

To isolate the semantic modulation effect, we add and subtract the term $\hat{\mathbf{v}}_n^b(\mathbf{x}_{t_n}, \mathrm{pt})$, which represents the unmodulated backward velocity evaluated at the same point $\mathbf{x}_{t_n}$:

$$\Delta\mathbf{x}_{n+1} = (\Delta t)^2 \big(([\hat{\mathbf{v}}_n^f(\mathbf{x}_{t_n}, \mathrm{pt}) - \hat{\mathbf{v}}_n^b(\mathbf{x}_{t_n}, \mathrm{ps}_n, \mathrm{pt})] + [\hat{\mathbf{v}}_n^b(\mathbf{x}_{t_n}, \mathrm{ps}_n, \mathrm{pt}) - \hat{\mathbf{v}}_{n+1}^b(\mathbf{x}_{t_{n+1}}, \mathrm{ps}_{n+1}, \mathrm{pt})])\big)^2 \tag{32}$$

Summing over all timesteps $n$ yields the final cumulative difference $\delta = \sum_n (\Delta\mathbf{x}_{n+1})^2$, completing the proof. $\qquad\square$

## B. Additional Experiments

To maintain a balance between computational efficiency and high-quality generation, our experimental evaluation focuses primarily on the Stable Diffusion 3.5 (SD3.5) architecture. While the FLUX model provides high-fidelity outputs, its substantial parameter count introduces significant memory overhead, making it less practical for the extensive comparative sampling required in this study. In contrast, SD3.5 serves as a more hardware-accessible and representative benchmark for rigorously evaluating diversity mechanisms without compromising on performance.

Since the official implementations of SPELL (Kirchhof et al., 2025), DiverseFlow (Morshed & Boddeti, 2025), and CNO (Kim et al., 2026) are currently unavailable, we re-implemented these baselines strictly adhering to the architectural configurations and hyperparameters specified in their respective publications. Qualitative and quantitative comparison results are presented in Figure 7.

*"a greeting card with a dress painted on it, on a dresser with various cosmetics"*

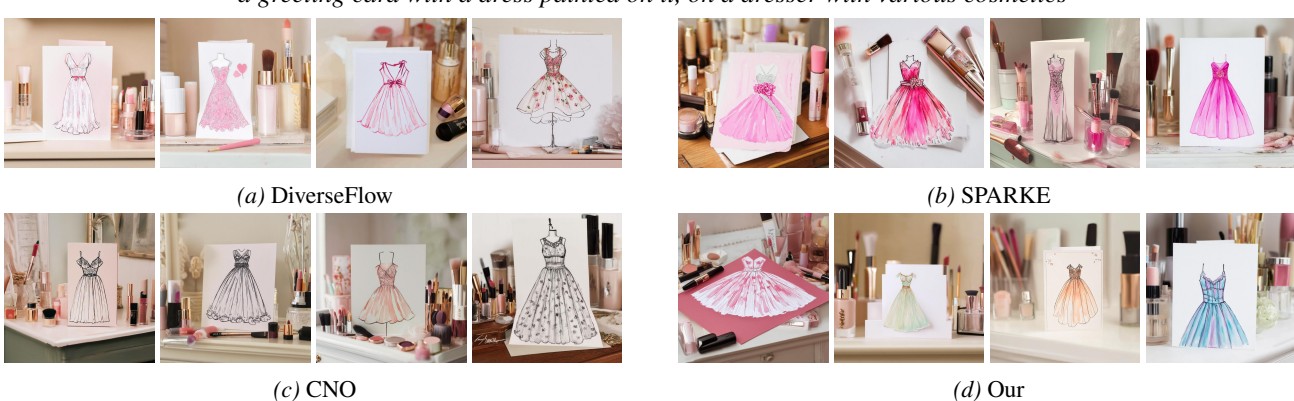

*(a)* DiverseFlow                 *(b)* SPARKE

*(c)* CNO                         *(d)* Our

*Figure 7.* Qualitative Comparison for Diversity Generation.

We evaluate two strategies for composing the stochastic prompts $\text{ps}_n$: (i) selecting random prompts from the MS-COCO dataset, and (ii) selecting arbitrary words from a general corpus. This comparison verifies the impact of semantic guidance on the diversity and stability of our stochastic prompt mechanism.

*"a model cal in a landscape"*

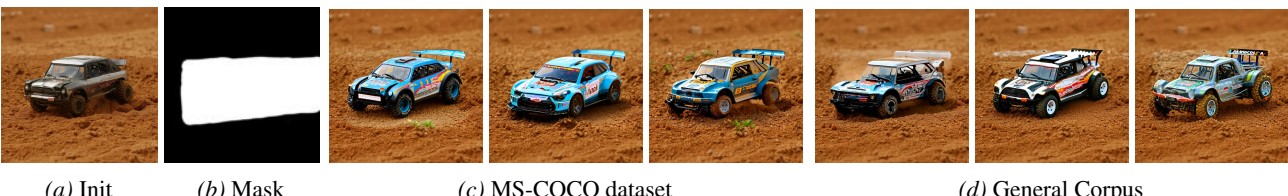

*(a)* Init        *(b)* Mask        *(c)* MS-COCO dataset                 *(d)* General Corpus

*Figure 8.* The Generation Result of Different Strategies for Composing $\text{ps}_n$.

Figure 9 illustrates the generation diversity under varying hyperparameter configurations. Notably, our framework exhibits a distinct advantage over Sparse Repellency (Morshed & Boddeti, 2025): while the latter suffers from negligible trajectory deviation when coefficients are small, our stochastic prompts sustain significant diversity even under minimal repulsive forces. This observed insensitivity to parameter scaling confirms that the stochastic velocity field serves as a robust, secondary mechanism for mode exploration, ensuring diverse sampling that is not solely reliant on repulsion magnitude. We emphasize that this stability holds even under a **minimal backward interval**, which inherently constrains the intensity of variation. Conversely, at higher coefficients (e.g., $\lambda = \zeta = 1$), the diversity is further amplified by the intensified repulsive forces, demonstrating that our stochastic mechanism works synergistically with existing joint sampling strategies.

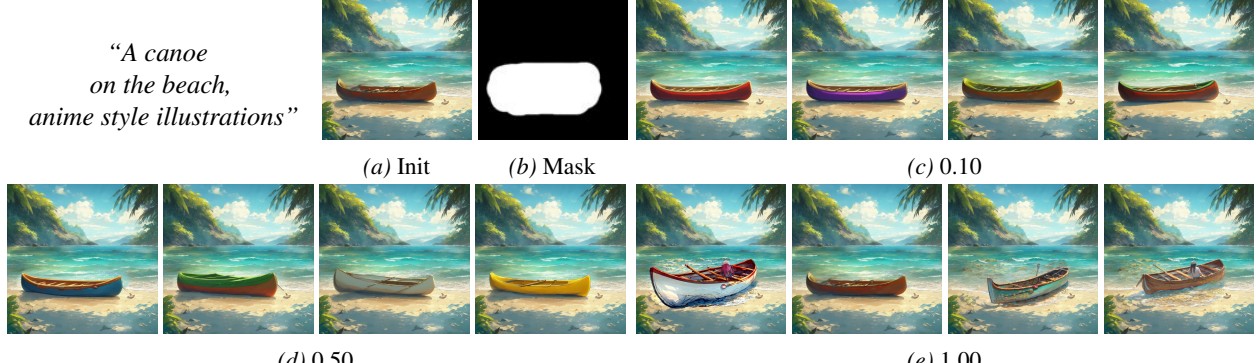

*Figure 9.* Generation result with Different $\lambda = \zeta$ Settings.

