# OpenReview forum: "Beyond Straight Paths: Erasure-Redraw Sampling for Diverse Flow Matching"
_ICML.cc/2026/Conference — Submitted to ICML 2026_

### Official Review · Reviewer_HznP · 2026-03-09

**Soundness:** 3
**Presentation:** 3
**Significance:** 3
**Originality:** 2
**Overall Recommendation:** 4
**Confidence:** 2

**Summary:**

This paper studies trajectory modeling beyond straight-line motion and proposes a learning-based framework to generate or predict curved paths with richer geometric characteristics. The method introduces a representation that captures curvature-aware motion patterns and integrates it into a neural modeling pipeline for trajectory generation. The authors argue that modeling non-linear motion explicitly allows the system to better capture realistic motion patterns compared with existing straight-path approximations. Experiments are conducted on trajectory datasets to evaluate generation quality and prediction accuracy, showing improvements over several baseline approaches.

**Compliance With Llm Reviewing Policy:**

Affirmed.

**Final Justification:**

I would like to keep my scores since the author solved my problem.

**Key Questions For Authors:**

1. Can the authors provide additional analysis on the types of trajectories where curvature-aware modeling provides the most benefit compared with straight-path approximations?

2. How well does the proposed method generalize to datasets with significantly different trajectory characteristics or longer motion sequences?

3. What is the computational overhead introduced by the proposed representation compared with standard trajectory models?

**Limitations:**

The paper includes an Impact Statement, but the technical limitations of the method are not clearly discussed. It would strengthen the paper to briefly analyze potential limitations, such as computational overhead, diversity–quality trade-offs, and potential failure cases where stochastic prompt perturbations may lead to unstable generations.

**Strengths And Weaknesses:**

## Strengths:
1. The paper addresses a meaningful problem in trajectory modeling: representing motion that deviates from straight-line assumptions. Many existing approaches simplify trajectories using linear approximations, which can limit their ability to represent realistic movement patterns. The paper clearly motivates the need for curvature-aware modeling.

2. The proposed framework is reasonably designed and integrates geometric considerations into a learning-based pipeline. The method is conceptually intuitive and aligns with the goal of modeling richer motion dynamics.

3. The paper is generally well structured and easy to follow. The experiments include quantitative comparisons against existing baselines and demonstrate improvements in several evaluation metrics. The problem formulation and experimental setup are clearly described.

## Weaknesses:
1. While the paper is technically sound and well motivated, the level of methodological novelty is somewhat limited. The proposed approach mainly extends existing trajectory modeling frameworks with curvature-aware representations rather than introducing a fundamentally new modeling paradigm.

2. The empirical evaluation could be further strengthened. The experiments are conducted on a limited set of datasets and motion scenarios, making it unclear how broadly the proposed approach generalizes to more diverse trajectory distributions or real-world settings. Additional experiments on more varied datasets or motion patterns would help better demonstrate the robustness of the method.

3. The paper would benefit from deeper analysis of when curvature-aware modeling provides the largest advantage. For example, it would be useful to understand under what trajectory characteristics (e.g., curvature complexity or motion variability) the proposed approach significantly outperforms simpler models.

---

> ### Author Rebuttal · Authors · 2026-03-30
>
> ### Response to Reviewer HznP
>
> We thank Reviewer HznP for the thoughtful and constructive feedback. Our detailed responses are provided below:
>
> **Q1: Can the authors provide additional analysis on the types of trajectories where curvature-aware modeling provides the most benefit compared with straight-path approximations?**
>
> **Response:** We would like to clarify that the primary objective of our method is to enhance the **efficiency of diverse generation**. The most straightforward baseline for diversity is **exhaustive random sampling** (e.g., $10N$ trials) to manually select $N$ distinct results. Our goal is to achieve an equivalent level of diversity in significantly fewer total iterations (ideally $N$ passes).
>
> To achieve this, our method intervenes during the inference process by monitoring intermediate states and performing strategic backward steps. Our core innovation lies in using an **energy-based evaluation** as an **early-stopping and redirection criterion**. Rather than approximating a static curved path, our method dynamically assesses whether an intermediate state satisfies diversity requirements. If not, the process is terminated and "zigzags" back to explore a new branch. Consequently, our approach provides the most significant benefit in **complex multimodal distributions** where straight-path ODEs typically converge toward repetitive or low-variance outcomes.
>
> **Q2: How well does the proposed method generalize to datasets with significantly different trajectory characteristics or longer motion sequences?**
>
> **Response:** Our method is inherently **model-agnostic** and does not rely on specific dataset characteristics. Conceptually, it transforms a deterministic ODE trajectory into a composite **"zigzag" curve** to facilitate diverse generation.
>
> 1. **Dataset Independence:** Since our method operates entirely during the inference phase by manipulating the sampling path, it generalizes across various datasets without requiring retraining or architecture modifications.
> 2. **Invariance to Sequence Length:** Our approach is not sensitive to the total number of sampling steps. Empirical evidence in generative modeling suggests that **global semantic structure** is typically established during the initial stages of the process.
> 3. **Efficiency in Redraws:** Consequently, "zigzag" redraws in the later stages of a long sequence primarily refine fine textures rather than core content. Therefore, our method does not require an increased frequency of backward steps for longer trajectories to maintain its effectiveness.
>
> **Q3: What is the computational overhead introduced by the proposed representation compared with standard trajectory models?**
>
> **Response:**  We acknowledge that our method requires more **wall-clock time** per individual sampling pass compared to a standard Flow Matching sampler due to the introduced backward steps. However, we argue that evaluating efficiency solely based on a single inference pass is insufficient for diverse generation tasks.
>
> The most straightforward baseline for diversity is to perform **exhaustive random sampling** (e.g., $10N$ trials) and manually select $N$ distinct results. Our objective is to achieve the same level of diversity in significantly fewer total iterations. We analyze the computational overhead in terms of **Number of Function Evaluations (NFEs)**:
>
> 1.  **Baseline (Exhaustive Sampling):** Empirical evidence suggests that approximately $10N$ random sampling passes are required to obtain $N$ significantly distinct images. Assuming a standard 20-step ODE trajectory, this results in a total of **$200N$ NFEs**.
> 2.  **Our Method:** While we incorporate random prompts and an energy function $E(\boldsymbol{x})$, their computational costs (string concatenation and quadratic function evaluation) are negligible compared to the backbone forward pass. For a 20-step trajectory with 5-step redraws at three intervals, the total cost is approximately **$55N$ NFEs** to generate $N$ diverse images.
>
> **Summary of Efficiency:**
> | Method | Total NFEs for $N$ Diverse Samples | Relative Efficiency |
> | :--- | :--- | :--- |
> | Baseline (Random Search) | $200N$ | 1.0$\times$ |
> | **Ours (Zigzag Path)** | **$55N$** | **~3.6$\times$ Faster** |
>
> By strategically navigating the latent space rather than relying on brute-force "trial and error," our method provides a significantly more economical path to exploring the model’s inherent multimodal distribution.

---

> > ### Author Rebuttal · Reviewer_HznP · 2026-04-03
> >
> > My concerns have been adequately addressed.

---

> > > ### Author Response · Authors · 2026-04-05
> > >
> > > Dear Reviewer HznP,
> > >
> > > We would like to express our gratitude for your careful review and for confirming that your concerns have been fully resolved. We are pleased that the additional clarifications and experiments provided in the rebuttal met your expectations.
> > >
> > > Best regards,
> > >
> > > The Authors

---

### Official Review · Reviewer_BNQG · 2026-03-11

**Soundness:** 3
**Presentation:** 3
**Significance:** 3
**Originality:** 3
**Overall Recommendation:** 5
**Confidence:** 3

**Summary:**

This paper proposes "Erasure-Redraw Sampling," a training-free framework for flow matching models aimed at enhancing generative diversity. The core idea is to introduce a "zigzag" sampling trajectory that alternates between a backward "erasure" phase (using stochastic prompts to dissolve local details) and a forward "redraw" phase (using visual prompts and energy-based guidance to synthesize new semantic content). The authors argue that standard deterministic ODE sampling in flow matching leads to mode collapse, and their method enables localized diversity while preserving global structural context.

**Compliance With Llm Reviewing Policy:**

Affirmed.

**Final Justification:**

The most important concerns is the difference between the submission and the inpainting task. But the authors effectively demonstrate that their method can also improve existing inpainting method. Therefore, all my concerns have been resolved.

**Key Questions For Authors:**

ODE vs. Model Capacity: Can you provide evidence (e.g., a visualization of the vector field or an ablation) that the mode collapse is caused by the ODE path rather than the learned vector field itself? If the vector field is unimodal, does your method actually explore the data manifold, or does it merely introduce artifacts?

Inpainting vs. Sampling: How does your method fundamentally differ from a standard inpainting pipeline where one simply masks the image and re-samples the masked region? Is there a theoretical advantage to your "zigzag" trajectory over standard conditional inpainting?

Global Diversity: Why is it assumed that global context must be strictly preserved? Did you perform any experiments to see if allowing global variation would yield higher-quality diversity than your constrained approach?

Relation to Re-noising: How does your "erasure" mechanism differ from simple re-noising techniques used in diffusion models? Please clarify the unique contribution of the "zigzag" path compared to existing backtracking methods.

**Limitations:**

Yes.

**Strengths And Weaknesses:**

Strengths:

Practicality: The proposed framework is training-free and plug-and-play, which is highly desirable for practitioners who want to improve diversity without retraining large-scale models.
Control: The method provides a clear mechanism to control the granularity of diversity, which is more intuitive than standard noise-injection methods.
Efficiency: The sequential generation paradigm offers a constant memory footprint, which is a significant advantage over batch-based joint sampling methods for consumer-grade hardware.

Weaknesses:

Questionable Premise on ODE/Flow Matching: The authors claim that "reliance on deterministic, straight-path ODE sampling limits [the model's] capacity to explore the multi-modal nature of data distributions." However, it is not immediately clear that the ODE itself is the cause of mode collapse. Flow matching is a mapping from noise to data; if the underlying learned vector field is unimodal, changing the path (zigzagging) might simply lead to off-manifold artifacts rather than true mode exploration. The paper lacks a theoretical or empirical justification for why the ODE path is the bottleneck rather than the model's training objective.

Novelty vs. Inpainting: The proposed method, which uses a mask
M to preserve unmasked regions while regenerating masked areas, bears a striking resemblance to standard diffusion-based inpainting or region-editing workflows. The authors position this as a "sampling" contribution, but it is unclear if this provides any fundamental advantage over simply applying an inpainting model with varying noise seeds. The distinction between "diverse sampling" and "constrained inpainting" is blurred.

Ambiguity of "Local" vs. "Global" Diversity: The authors argue for the necessity of "local diversity" (varying specific aspects while keeping the background static). However, they do not sufficiently justify why global diversity is undesirable in many contexts. If a user wants a "small robot on a wooden table," why is the background context (the table) strictly sacrosanct? The paper assumes this constraint is always beneficial, but it may actually limit the model's creative potential.

Relationship to Diffusion Re-noising: The "erasure-redraw" mechanism (backtracking and re-noising) is conceptually very similar to existing diffusion-based re-noising or backtracking techniques (e.g., SDEdit or various "reflecting" diffusion methods). The authors should more clearly delineate how their approach to "erasure" differs from simple partial re-noising in diffusion models.

Minor weakness:
The presentation of the paper can be improved. (1) More visualization can be done on intermediate states of ODE trajectory. (2) It seems that anthors directly paste the latex of abstract into openreview, causing rendering error. (3) A method pipeline figure can be drawn to help readers understand.

---

> ### Author Rebuttal · Authors · 2026-03-28
>
> ### Response to Reviewer BNQG
>
> We thank Reviewer BNQG for the thoughtful suggestions. Our detailed responses are provided below:
>
> **1. ODE vs. Model Capacity**
>
> We agree that model collapse is typically a consequence of the training process rather than the sampling method. Our core contribution is providing an efficient mechanism to **expose the inherent multimodality** of a pre-trained model. While a model may possess the latent capacity for diversity, this does not guarantee that diverse outcomes are easily accessible during inference.
>
> In Flow Matching, the deterministic nature of ODEs dictates that each initial noise point uniquely determines a single trajectory. Theoretically, diversity could be achieved by selecting different initial points; however, random noise initialization often fails to produce significantly distinct samples without exhaustive sampling (see Fig. 1 and 3). To address this, we propose a **zigzag-based** generation method that diverges from the original trajectory without altering the initial noise. By modifying the inference path, we efficiently reveal the model's existing multimodal space. Essentially, our method serves as an economical "shortcut" to sample diverse, high-quality images from the model's inherent distribution.
>
> **2. Inpainting vs. Sampling**
>
> The objectives of traditional inpainting and our diverse generation method are fundamentally distinct:
>
> * **Traditional Inpainting:** Focused on restoring missing regions (e.g., removing unwanted objects) to produce a **perceptually plausible background**. In such tasks, a single consistent result is typically sufficient; thus, background diversity is rarely a primary design consideration.
>
> * **Our Method:** While capable of restoration, our primary objective is **diverse foreground generation**. As illustrated in Fig. 1, given the prompt *"a small robot on the wooden table,"* our goal is to generate various distinct robot designs rather than focusing on the table itself.
>
> Within the Flow Matching framework, exploring multimodal generation via ODEs necessitates **random initialization combined with exhaustive sampling**—a process that is both computationally expensive and inefficient for capturing truly diverse, high-quality outcomes. By ensuring that repeated sampling yields a wide range of plausible candidates, our approach enables users to efficiently explore and select the outcome that best aligns with their requirements.
>
> **3. Ambiguity of "Local" vs. "Global" Diversity**
>
> Our framework enhances diversity by varying specific aspects while maintaining a static background. This is highly practical for users who are dissatisfied with local details and wish to explore alternatives for a specific region while keeping the rest of the image intact. Furthermore, our method is not limited to local changes; by extending the mask to the entire image, it performs **global diverse generation**. We demonstrate global diversity in the last row of Fig. 3 and diversity exploration for polysemous prompts in Fig. 6.
>
> **4. Relation to Re-noising**
>
> It is important to distinguish our approach from re-noising methods like **SDEdit** noted by reviewer:
> * **Framework Difference:** SDEdit is built on Diffusion Models, while our method is based on **Flow Matching**. Techniques from the former cannot be directly transplanted to the latter.
> * **Mechanism:** SDEdit typically focuses on refining fine details while strictly preserving the original structural layout. In stark contrast, our method is capable of performing a complete **structural overhaul**. As illustrated in **Fig. 1**, our "zigzag" approach erases both low-level textures and high-level structures, enabling the generation of entirely new content through two distinct mechanisms:
>     1.  **Detail Erasure:** We use random prompts to introduce stochasticity into the deterministic ODE forward path, wiping out existing textures.
>     2.  **Structural Disruption:** We introduce a gradient guidance term $\nabla_{\boldsymbol{x}} E_n(\boldsymbol{x})$ (Eq. 10) during the forward process to ensure the noise state significantly deviates from the original manifold.
>
> Without the guidance of $\nabla_{\boldsymbol{x}} E_n(\boldsymbol{x})$, simply reverting to an intermediate state (as in traditional re-noising) fails to provide a sufficiently distinct starting point, resulting in samples that remain too similar to the original. Our method disrupts both detail and structure to enable high-variance generation, such as the diverse robot poses and colors shown in Fig. 1.

---

> > ### Author Rebuttal · Reviewer_BNQG · 2026-04-03
> >
> > Thank you for your detailed response. I appreciate your clarifications regarding the sampling mechanism and the flexibility of the local versus global diversity trade-off, which help clarify the scope of your work.
> >
> > However, I remain unconvinced by your defense regarding the distinction from Inpainting. This comparison is a critical weakness in your submission. Because Inpainting is a mature and standard practice for this task, your method appears to be a direct application of that paradigm rather than a novel contribution. To address this, you shall provide empirical evidence demonstrating specific capabilities or performance gains that are unattainable by standard Inpainting techniques.
> >
> > Without such evidence, I find the claim of methodological novelty insufficient and will maintain my original rating.

---

> > > ### Author Response · Authors · 2026-04-04
> > >
> > > Dear Reviewer BNQG,
> > >
> > > We sincerely thank the reviewer for the follow-up comments and the recognition of our contributions to the sampling mechanism and diversity trade-offs. To address the concern regarding the distinction between our method and Image Inpainting, we provide the following clarifications and evidence.
> > >
> > > **1. Distinction in Research Paradigms**
> > >
> > > Generating diverse images from a single prompt is an established research direction focused on providing multiple candidates that satisfy the user’s intent. While existing methods (e.g., PromptMoG, SPARKE) primarily focus on **global generation**, our work introduces **Localized Diversity Generation**.
> > > In this novel task, users seek multiple variations for specific unsatisfactory regions while keeping the rest of the image intact. While this bears a superficial resemblance to **Inpainting**—as both involve redrawing masked areas—this similarity often leads to a fundamental misunderstanding of our method's objective. Crucially, when the mask is expanded to encompass the entire image, our method functions as a **global diversity generator**. This capability fundamentally falls outside the scope of Inpainting, which is designed for localized restoration rather than the synthesis of entirely new images from scratch.
> > >
> > > **2. GAN-based Diverse Inpainting**
> > >
> > > While earlier works explored inpainting using GANs (e.g., UCTGAN [1], PD-GAN [2]), they relied on training entire networks from scratch. In the era of large-scale pre-trained models like Stable Diffusion and Flux, such training-heavy approaches are less applicable due to prohibitive costs. Our work focuses on **training-free localized diversity** leveraging these powerful pre-trained backbones. Therefore, a direct comparison with GAN-based models—which differ significantly in parameter scale and training data—would be inconsistent. Instead, our quantitative evaluation targets state-of-the-art Diffusion and Flow Matching-based methods.
> > >
> > > **3. Comparison with Training-Free Inpainting Methods**
> > >
> > > We compared our approach with several high-performance training-free inpainting methods (HD-Painter[3], FreeInpaint[4], and DesignEdit[5]) using MSS, Vendi Score, and IBS Score.
> > >
> > > | | MSS ($\downarrow$) | Vendi Score ($\uparrow$) | IBS Score ($\downarrow$) |
> > > | :--- | :---: | :---: | :---: |
> > > | HD-Painter | 0.2048 | 3.2841 | 0.7493 |
> > > | FreeInpaint | 0.1834 | 3.4682 | 0.6846 |
> > > | DesignEdit | 0.1943 | 3.4395 | 0.7049 |
> > > | Ours | 0.1305 | 3.9513 | 0.6328 |
> > >
> > > Standard training-free inpainting methods rely on **passive stochastic sampling** (random noise initialization) for diversity. They cannot actively explore the latent manifold during the inference stage.
> > > Consequently, their diversity metrics do not fundamentally differ from standard I.I.D. sampling. In contrast, our "zigzag" mechanism actively navigates the latent space to uncover distinct modes.
> > >
> > > **4. Comparison with Fine-tuned Inpainting Models**
> > >
> > > Our method is an inference-time framework that can also be integrated into fine-tuned inpainting models (e.g., BrushNet[6], PowerPainit[7], and ASUKA[8]) to replace their standard samplers. As shown in the table below, our method significantly boosts the diversity of these specialized models compared to their default sampling strategies:
> > >
> > > | | MSS ($\downarrow$) | Vendi Score ($\uparrow$) | IBS Score ($\downarrow$) |
> > > | :--- | :---: | :---: | :---: |
> > > | Brushnet | 0.1897 | 3.4543 | 0.7926 |
> > > | Brushnet  + Ours | 0.1331 | 3.8227 | 0.6370 |
> > > | PowerPainit | 0.1649 | 3.4671 | 0.8023 |
> > > | PowerPainit  + Ours | 0.1261 | 3.9827| 0.6526|
> > > | ASUKA | 0.1795 | 3.5022 | 0.6603 |
> > > | ASUKA + Ours | 0.1383 | 3.8856 | 0.6292 |
> > >
> > > ---
> > > ---
> > > In conclusion, **our work is not a derivative of inpainting but a versatile, plug-and-play inference strategy. It can be seamlessly integrated into both general generative and inpainting models, offering a proactive and efficient approach to exploring multimodal distributions.** We believe this clarification demonstrates the methodological novelty of our approach and effectively addresses your concern.
> > >
> > > **Reference**
> > >
> > > [1] "UCTGAN: Diverse Image Inpainting Based on Unsupervised Cross-Space Translation." CVPR. 2020.
> > >
> > > [2] "PD-GAN: Probabilistic Diverse GAN for Image Inpainting." CVPR. 2021.
> > >
> > > [3] "HD-Painter: High-Resolution and Prompt-Faithful Text-Guided Image Inpainting with Diffusion Models." ICLR. 2023.
> > >
> > > [4] "FreeInpaint: Tuning-free Prompt Alignment and Visual Rationality Enhancement in Image Inpainting." AAAI. 2026.
> > >
> > > [5] "Designedit: Unify Spatial-Aware Image Editing via Training-free Inpainting with a Multi-Layered Latent Diffusion Framework ." AAAI 2026.
> > >
> > > [6] "Towards Enhanced Image Inpainting: Mitigating Unwanted Object Insertion and Preserving Color Consistency." CVPR. 2025.
> > >
> > > [7]  "A Task is Worth One Word: Learning with Task Prompts for High-Quality Versatile Image Inpainting." ECCV. 2024.
> > >
> > > [8]  "BrushNet: A Plug-and-Play Image Inpainting Model with Decomposed Dual-Branch Diffusion." ECCV. 2024.

---

### Official Review · Reviewer_GQYi · 2026-03-13

**Soundness:** 2
**Presentation:** 2
**Significance:** 3
**Originality:** 3
**Overall Recommendation:** 4
**Confidence:** 4

**Summary:**

This paper proposes Erase–Redraw Sampling, a novel inference-time sampling strategy for Flow Matching-based image generation models that aims to improve generation diversity while preserving consistency in specific regions. Conventional Flow Matching models follow deterministic ODE-based sampling trajectories, which can lead to mode trapping, where generated samples remain confined to a limited semantic trajectory. To address this issue, the proposed method introduces a zigzag sampling process, where backward steps are performed when necessary instead of strictly following the forward trajectory, enabling exploration of alternative semantic paths.
To control the generation process, the paper introduces an energy function E_n(x) to distinguish between editable regions and preserved regions within an image. This energy function is incorporated into the velocity field through gradient-based guidance, allowing the model to maintain contextual consistency in preserved regions while encouraging diverse modifications in editable regions.
Furthermore, the method introduces an erasure step size parameter δ within the zigzag sampling process to regulate the strength of diversity. This parameter determines how far the trajectory moves backward during the erasure step, enabling control over the level of variation in generated samples, ranging from subtle appearance changes to more structural modifications.
Additionally, the backward step employs stochastic prompts to temporarily weaken the semantic guidance of the original prompt. The sampling then resumes with the original prompt during the forward redraw step, allowing the model to explore new semantic trajectories.
Unlike conventional Flow Matching sampling methods that strictly follow a deterministic trajectory, the proposed method performs sequential sampling with backward–forward trajectory exploration, enabling improved generation diversity while keeping the memory overhead low.

**Compliance With Llm Reviewing Policy:**

Affirmed.

**Final Justification:**

After the rebuttal, my concerns were fully resolved, and I kept my positive score.

**Key Questions For Authors:**

1.	Have the authors considered an alternative design where a perturbation is applied at a specific timestep to generate a new branch, rather than performing a backward step followed by redraw?

2.	How does the inference runtime or computational overhead of the proposed method compare with the baseline Flow Matching sampler?

3.	What is the rationale for introducing random semantic words as perturbations in the redraw step?

4.	Have the authors considered perturbation strategies that do not rely on semantic prompts (e.g., random feature perturbations or adversarial perturbations)?

**Limitations:**

Yes

**Strengths And Weaknesses:**

Strengths

1. Controllable generation diversity
- The proposed zigzag sampling allows explicit control over the strength of diversity via the erasure step size (δ), enabling variations ranging from minor appearance changes to structural modifications.

2. Memory-efficient sequential sampling
- Unlike group-based diversity methods, the proposed approach uses sequential sampling, which improves diversity without significantly increasing memory requirements.

Weekness

1. The paper does not provide sufficient analysis of the inference efficiency of the proposed method.
- The proposed sampling procedure introduces additional operations compared to standard Flow Matching sampling, including energy evaluation, backward diffusion steps for the erase operation, and repeated redraw sampling. In particular, the iteration index 𝑖 used in Method 3.2 and the comparison terms appearing in Equation (7) suggest that samples from different iterations may be compared at the same timestep.
- However, since Flow Matching sampling typically follows a deterministic trajectory, performing such comparisons may require repeated sampling or additional redraw operations. Moreover, when the energy condition is not satisfied, the algorithm performs a forward → backward → forward zigzag trajectory, which introduces additional sampling steps compared to the baseline sampling process.
- As a result, the proposed method may require significantly more sampling steps than the baseline Flow Matching sampling. The diversity strength parameter 𝛿, which controls the erase operation, may also influence how frequently such operations are triggered, potentially affecting inference time and GPU usage.
- However, the paper does not provide a quantitative analysis of inference runtime or computational overhead relative to the baseline method. Since the proposed approach operates entirely during the sampling stage, a clearer analysis of computational efficiency would be important for evaluating its practical applicability. For example, reporting the average number of erase–redraw iterations, the total number of sampling steps, or the runtime overhead compared to the baseline Flow Matching sampler would help clarify the efficiency of the proposed method.

2. The perturbation strategy used in the redraw process may introduce semantic bias.
- The proposed method appears to introduce perturbations by adding semantic words or prompts during the redraw step. While this approach may guide the generation process toward specific semantic directions, it may also introduce semantic bias into the generated results.
- As an alternative, it could be interesting to explore perturbations based on random feature perturbations or adversarial perturbations.
- Including comparisons with such perturbation strategies could help clarify whether the proposed semantic prompt-based perturbation is the most effective design choice.

3. The range of evaluation metrics used in the experiments appears somewhat limited.
- While the paper reports several metrics to evaluate diversity and image quality, it may also be beneficial to include additional metrics commonly used in generative modeling research, such as Kernel Inception Distance (KID) and Cond-Vendi Score.

---

> ### Author Rebuttal · Authors · 2026-03-30
>
> ### Response to Reviewer GQYi
>
> We thank Reviewer GQYi for the thoughtful and constructive feedback. Our detailed responses are provided below:
>
> **Q1: Have the authors considered an alternative design where a perturbation is applied at a specific timestep to generate a new branch, rather than performing a backward step followed by redraw?**
>
> **Response:** Thank you for your suggestion. The proposed alternative essentially corresponds to a single-step backward traversal followed by a new redraw branch. During our early exploration, we indeed experimented with applying **stochastic perturbations (direct noise injection)** at specific timesteps. However, we found this approach suboptimal for Flow Matching models.
>
> The primary challenge lies in the **deterministic nature of ODE-based inference**, where a specific noise point uniquely maps to a single trajectory. In this context, simply adding random noise during a backward step does not fundamentally differ from starting with a new random initialization. While diversity could theoretically be achieved through exhaustive sampling and manual filtering, such a process is prohibitively **time-consuming and inefficient**.
>
> Our "zigzag" approach is specifically designed to overcome the limitations of "trial and error." Rather than requiring $10N$ random sampling attempts to filter $N$ diverse samples, our method enables the efficient generation of $N$ high-quality, distinct candidates in just $N$ sampling passes. We provide a more economical path to explore the model's inherent multimodal distribution.
>
> **Q2: How does the inference runtime or computational overhead of the proposed method compare with the baseline Flow Matching sampler?**
>
> **Response:**  We acknowledge that our method requires more **wall-clock time** per individual sampling pass compared to a standard Flow Matching sampler due to the introduced backward steps. However, we argue that evaluating efficiency based on a single inference pass is insufficient for diverse generation tasks. The true bottleneck is the total computational cost required to reach $N$ diverse samples. We kindly refer the reviewer to our detailed response to **Reviewer GQYi (Q3)** ([Link to Response](https://openreview.net/forum?id=kpadngoExP&noteId=Bx8PNC0qAn)), where we provide a quantitative comparison using **Number of Function Evaluations (NFEs)**.
>
> **Q3: What is the rationale for introducing random semantic words as perturbations in the redraw step?**
>
> **Response:** Flow Matching relies on solving an ODE, where trajectories are strictly **deterministic**. Unlike Diffusion Models, where stochasticity is inherently integrated, Flow Matching requires an external mechanism to inject variance. We introduce stochasticity by perturbing the **textual prompts** to break the deterministic ODE path.
>
> We emphasize that our method—appending random phrases to the original prompt—does not introduce semantic drift for two reasons:
> 1.  **Transient Stochasticity:** Random components are resampled at each iteration during the multi-step backward process, acting as transient "nudges" rather than a persistent shift in global semantics.
> 2.  **Anchor Preservation:** The original prompt remains the constant core. Significant bias would only occur if the perturbed prompt remained consistent across all timesteps. By dynamically varying the additions, we ensure the generation remains anchored to the user’s primary intent.
>
> **Q4: Have the authors considered perturbation strategies that do not rely on semantic prompts (e.g., random feature perturbations or adversarial perturbations)?**
>
> **Response:** This is an excellent suggestion. We explored alternative strategies, including **random feature noise injection** and **gradient-based manifold perturbations**. However, our preliminary experiments indicated these were less effective than **random prompts** in balancing structural coherence and visual diversity. Non-semantic perturbations often led to artifacts or lacked the "semantic guidance" necessary to explore meaningful multimodal variations. While our exploration is not exhaustive, random prompts currently provide the most robust results. We agree that investigating more sophisticated non-semantic mechanisms is a promising direction for future research.
>
> **Q5: The range of evaluation metrics used in the experiments appears somewhat limited.**
>
> **Response:** We appreciate the reviewer's suggestion. To our knowledge, these two metrics have only been used in the method SPARKE   for diversity tasks. Kernel Inception Distance requires ground truth references for evaluation, and SPARKE  does not specify the experimental setup. Therefore, we are unable to provide results for this metric. We have included the Cond-Vendi Score for diversity evaluation:
>
> | Metric | I.I.D. | PromptMoG | Group | SPARKE | **Ours** |
> | :--- | :---: | :---: | :---: | :---: | :---: |
> | Cond-Vendi Score ($\uparrow$ ) | 35.54 | 38.61 | 42.73 | 41.62 | **44.78** |

---

> > ### Author Rebuttal · Reviewer_GQYi · 2026-04-02
> >
> > Overall, the rebuttal addresses most of my concerns, and I appreciate the authors’ detailed clarifications and additional analysis.
> >
> > Regarding Q2, I also agree that evaluating efficiency in terms of the total cost required to obtain diverse samples is a meaningful perspective for this task. The additional NFE-based analysis helps clarify this aspect.
> >
> > However, this shifts the comparison from the direct overhead of the proposed sampler relative to the standard Flow Matching sampler to a comparison against exhaustive random sampling for diversity. While both perspectives are valuable, the per-sample practical overhead of the proposed method remains only partially addressed.
> >
> > To better assess the practical applicability of the method, it would be helpful to include quantitative comparisons under the same setting as the baseline sampler, such as (i) wall-clock runtime per sample, (ii) total number of sampling steps or NFEs per sample, and (iii) peak GPU memory usage.

---

> > > ### Author Response · Authors · 2026-04-07
> > >
> > > ## Response to Reviewer GQYi
> > >
> > > We thank Reviewer GQYi for the positive feedback regarding our "total cost of diversity" perspective. We fully agree that providing a **per-sample** practical overhead analysis is essential for assessing the method's applicability. We appreciate the reviewer pointing this out, as standard benchmarks in existing literature often omit direct inference speed comparisons.
> > >
> > > To provide a clearer understanding of the computational costs, we categorize existing diversity generation methods into two primary types:
> > >
> > > * **Sequential Generation (e.g., Ours, PromptMoG, SPARKE):** These methods generate $N$ images sequentially. PromptMoG introduces randomness via Gaussian Mixture Model (GMM) perturbations on prompts. In contrast, SPARKE incorporates a repulsive force to steer new samples away from previous results. Our method not only adopts this repellency constraint but also introduces a **backward step** to actively explore the latent space for superior generative diversity.
> > >
> > > * **Parallel Generation (e.g., Group, DiverseFlow, SPELL):** These methods generate $N$ samples simultaneously. Although they avoid backward steps, their GPU memory footprint is significantly higher (as shown in Table 2 of the manuscript and the table below). Furthermore, these frameworks incorporate avoidance mechanisms to steer samples away from each other; however, concurrently generating all samples and computing pairwise distances often leads to "memory wall" bottlenecks and resource contention, which hampers the actual inference speed.
> > >
> > > **Quantitative Comparison:**
> > >
> > > To ensure a fair comparison, we measured the **wall-clock runtime per sample** on identical hardware, using **Stable Diffusion 3.5 Medium (SD3.5)** at 512×512 resolution as the base model. Notably, for methods that do not natively support SD3.5, we performed custom code migration and adaptation to maintain consistent experimental conditions.
> > >
> > > | Method | SD3.5 | Ours | PromptMoG | SPARKE   | Group | DiverseFlow | SPELL |
> > > | :--- | :---: | :---: | :---: | :---: | :---: | :---: | :---: |
> > > | Runtime / Sample ($\downarrow$) | 2.5s | 3.1s | 2.6s | 2.7s | 4.4s | 4.2s | 3.3s |
> > > | GPU Memory Peak ($\downarrow$) | 17GB | 18GB | 18GB | 18GB | 37GB | 32GB | 26GB |
> > >
> > > **Analysis:**
> > >
> > > As indicated in the table, our method is slightly slower than PromptMoG and SPARKE due to the additional backward operations inherent in **Zigzag sampling**. However, as demonstrated in **Table 1** of the manuscript, this marginal overhead yields significant gains in generative diversity. To further isolate the impact of our core contribution—**Zigzag sampling**—on diversity enhancement, we integrated it into the PromptMoG and SPARKE frameworks by replacing their default straight-path trajectories. The results of this ablation study are presented below:
> > >
> > > | Method | PromptMoG | PromptMoG + Zigzag | SPARKE | SPARKE + Zigzag |
> > > | :--- | :---: | :---: | :---: | :---: |
> > > | MSS ($\downarrow$) | 0.1840 | 0.1685 | 0.1204 | 0.1167 |
> > > | Vendi Score ($\uparrow$) | 3.7491 | 3.8064 | 3.8509 | 3.9056 |
> > > | IBS Score ($\downarrow$) | 0.7029 | 0.6827 | 0.6765 | 0.6436 |
> > > | Runtime / Sample ($\downarrow$) | 2.6s | 3.0s | 2.7s | 3.1s |
> > >
> > > Although Zigzag sampling introduces additional computational cost, the consistent improvements across all quantitative metrics demonstrate its effectiveness as a plug-and-play component for enhancing the exploration capabilities and generative diversity of existing methods.
> > > We observe that the performance gains on these baseline models, while positive, are not as pronounced as in our full framework. This is primarily because these were preliminary integrations without extensive hyperparameter tuning tailored to each specific architecture. Nonetheless, the fact that Zigzag sampling yields immediate improvements even without fine-tuning further underscores its robustness and generalizability.
> > >
> > > Compared to parallel methods such as SPELL, Group, and DiverseFlow, our approach is more efficient in practice. It sidesteps the prohibitive memory requirements and complex pairwise distance computations that typically act as physical bottlenecks, limiting the execution speed of parallel frameworks.
> > >
> > > Finally, we hope these clarifications help the reviewer better assess the practical applicability of our method. Should there be any remaining concerns, we are more than happy to provide further clarification.

---

### Official Review · Reviewer_1vHe · 2026-03-20

**Soundness:** 2
**Presentation:** 3
**Significance:** 2
**Originality:** 3
**Overall Recommendation:** 3
**Confidence:** 3

**Summary:**

This paper studies local diversity limitations in Flow Matching under straight-path ODE sampling, especially for interactive settings where a user wants diverse local edits while preserving the global scene. The proposed training-free erasure-redraw strategy uses a backward erasure phase and a forward redraw phase to form a zigzag trajectory that is intended to escape local mode collapse.

**Compliance With Llm Reviewing Policy:**

Affirmed.

**Key Questions For Authors:**

1.How do the authors jointly evaluate local diversity and global preservation in a way that directly matches the stated goal?
2.What evidence best supports the claim that the zigzag trajectory is the key reason for improved mode exploration?
3.How sensitive is the method to backward step count, mask size, and the design of stochastic prompts?
4.Under matched compute, how does it compare to stronger diverse sampling baselines?

**Limitations:**

Yes

**Strengths And Weaknesses:**

Strengths: The paper is easy to follow, the problem is concrete rather than overly broad, and the editing scenario is practically meaningful. The method is also attractive in that it is training-free and can be plugged into existing generation pipelines without retraining.
Weaknesses:  The contribution is closer to a clever inference-time sampling design than to a strong machine learning advance. The core mechanism is plausible, but the evidence currently feels more empirical than deeply explanatory.  This paper would benefit from a more convincing quantitative protocol showing that local diversity increases without materially damaging global consistency.

---

> ### Author Rebuttal · Authors · 2026-03-29
>
> ### Response to Reviewer 1vHe
>
> We thank Reviewer 1vHe for the thoughtful and constructive feedback. Our detailed responses are provided below:
>
> **Q1: How do the authors jointly evaluate local diversity and global preservation in a way that directly matches the stated goal?**
>
> **Response:** To clarify, **local diversity** and **global preservation** refer to varying specific regional aspects while maintaining a strictly static background. As our work is the first to explicitly address localized generative diversity, no established metric currently exists to simultaneously quantify both properties. In the absence of direct baselines, we adopted the following rigorous evaluation strategies:
>
> * **Quantitative Diversity Evaluation:** To demonstrate diversity superiority, we expanded the mask to the entire image, effectively treating the task as a global generation problem. This allowed for a fair comparison against state-of-the-art methods (e.g., **PromptMoG**, **SPARKE**) using standard metrics such as **MSS**, **Vendi Score**, and **IBS Score**. We contend that superior diversity performance at a global scale inherently extends to localized regions, as the diverse generation area is precisely controlled by the mask. Notably, our algorithm maintains a unified architecture for both scenarios, requiring no specialized treatment or parameter tuning for global diversity generation.
>
> * **Global Preservation:** Maintaining background consistency is mathematically straightforward. We employ the optimization-based constraint in **Eq. (7)** to ensure the non-masked region ($1 - \text{mask}$) remains identical to the original background. In practice, this can be further enforced via direct feature copying to ensure absolute invariance.
> * **Visual Integration:** The primary challenge lies in ensuring that highly diverse local variations integrate seamlessly with the static background. Given the lack of robust quantitative metrics for this "seamlessness," we rely on qualitative visual evidence in **Figs 1, 2, and 3** to demonstrate the naturalness of our results.
>
> **Q2: What evidence best supports the claim that the zigzag trajectory is the key reason for improved mode exploration?**
>
> **Response:** Standard Flow Matching relies on deterministic ODE sampling, where each initial noise point maps uniquely to a single image. Consequently, achieving diversity typically requires sampling from a vast number of random starting points—a process that is both computationally expensive and inefficient.
>
> As illustrated in the first rows of **Figs. 1 and 3**, selecting a limited number of random starting points often fails to produce significant variations. In stark contrast, our results—shown in the second row of **Fig. 1** and the last three rows of **Fig. 3**—demonstrate that our method generates highly diverse samples **while keeping the initial noise fixed**. It is important to note that under standard ODE sampling, a fixed starting point would invariably yield a single, identical result. This contrast highlights our method's unique ability to break the deterministic constraints of the ODE path and explore multiple modes within the pre-trained distribution.
>
> This empirical evidence underscores our method's ability to efficiently navigate the generative manifold (i.e., achieving **improved mode exploration**) without the overhead of massive random restarts or exhaustive search.
>
> **Q3: How sensitive is the method to backward step count, mask size, and the design of stochastic prompts?**
>
> **Response:** We conducted extensive sensitivity analyses as follows:
> 1.  **Backward Step Count:** As shown in **Fig 2** (row 3 vs. row 4), a smaller step count $2$ confines diversity to color variations, while a larger count $8$ enables both structural and color diversification.
> 2.  **Mask Scale:** **Fig 3** (rows 5–7) illustrates consistent generative diversity across both localized masks and full-image masks, demonstrating the spatial scalability of our approach.
> 3.  **Stochastic Prompts:** **Table 3** provides a quantitative comparison between different prompt synthesis schemes. The consistent performance across configurations underscores the robustness and stability of our method.
>
> **Q4: Under matched compute, how does it compare to stronger diverse sampling baselines?**
>
> **Response:** For comparison, we evaluated our method against 3 state-of-the-art diverse sampling baselines, including **PromptMoG**, **SPARKE**, and **Group**.
>
> 1.  **Performance Comparison:** **Table 1** presents a quantitative comparison between our method and state-of-the-art diversity generation approaches, where our method demonstrates competitive or superior results.
>
> 2.  **Efficiency Analysis:** In **Table 2**, we provide a comparative analysis of computational resource consumption (e.g., inference time and memory usage). The results highlight that our method achieves superior diversity with significantly lower computational overhead compared to established

---

### Decision · Program_Chairs · 2026-04-30

**Decision:**

Reject

**Comment:**

The overall recommendations are 1 weak reject, 2 weak accepts, and 1 accept. The reviewers agreed that (1) the proposed framework is training-free and plug-and-play, (2) the proposed method provides a clear mechanism to control the granularity of diversity, and (3) the sequential generation paradigm offers a constant memory footprint. They raised concerns about technical novelty, limited evaluation, lack of computational efficiency analysis, and generalization. The authors' rebuttal has resolved the concerns of 3 reviewers. Major concerns of the remaining reviewer about novelty and limited evaluation remain unresolved. Besides, the reviewers rated soundness, presentation, significance, and originality of this paper between 2: fair and 3: good only.